# ZMYND8 mediated liquid condensates spatiotemporally decommission the latent super-enhancers during macrophage polarization

Pan Jia[1,4], Xiang Li[1,2,4], Xuelei Wang [1,4], Liangjiao Yao[1], Yingying Xu[1], Yu Hu[1], Wenwen Xu[1], Zhe He[1], Qifan Zhao[1], Yicong Deng[1], Yi Zang[1], Meiyu Zhang[3], Yan Zhang[3✉], Jun Qin [1,2✉] & Wei Lu [1✉]

Super-enhancers (SEs) govern macrophage polarization and function. However, the mechanism underlying the signal-dependent latent SEs remodeling in macrophages remains largely undefined. Here we show that the epigenetic reader ZMYND8 forms liquid compartments with NF-κB/p65 to silence latent SEs and restrict macrophage-mediated inflammation. Mechanistically, the fusion of ZMYND8 and p65 liquid condensates is reinforced by signal-induced acetylation of p65. Then acetylated p65 guides the ZMYND8 redistribution onto latent SEs de novo generated in polarized macrophages, and consequently, recruit LSD1 to decommission latent SEs. The liquidity characteristic of ZMYND8 is critical for its regulatory effect since mutations coagulating ZMYND8 into solid compartments disable the translocation of ZMYND8 and its suppressive function. Thereby, ZMYND8 serves as a molecular rheostat to switch off latent SEs and control the magnitude of the immune response. Meanwhile, we propose a phase separation model by which the latent SEs are fine-tuned in a spatiotemporal manner.

[1] CAS Key Laboratory of Tissue Microenvironment and Tumor, Shanghai Institute of Nutrition and Health, University of Chinese Academy of Sciences, Chinese Academy of Sciences, Shanghai, China. [2] CAS Center for Excellence in Molecular Cell Science, Chinese Academy of Sciences, Shanghai, China. [3] Shanghai Institute of Immunology, Department of Immunology and Microbiology, State Key Laboratory of Oncogenes and Related Genes, Shanghai Jiao Tong University School of Medicine, Shanghai Jiao Tong University School of Medicine, Shanghai 200025, China. [4] These authors contributed equally: Pan Jia, Xiang Li, Xuelei Wang. ✉email: zhy3331a@shsmu.edu.cn; qinjun@sibs.ac.cn; lvwei@sibs.ac.cn

Macrophages are highly plastic and differentiate into distinct subsets, which requires epigenetic reprogramming to establish a lineage-specific enhancer profile[1,2]. Besides constitutively activated enhancers, primed (poised) enhancers are pre-occupied by lineage-determining transcription factors (LDTF) PU.1 and/or C/EBP during macrophage development. While some signal-dependent transcription factors (SDTF) can open a set of previously packed genomic regions to generate latent (de novo) enhancers[3,4]. Both primed and latent enhancers trigger specific gene transcription in response to stimulations[3–6]. The preserved epigenetic modifications on latent enhancers contribute to the memory of innate immune response, also known as trained immunity[3].

Recently some massive clusters of *cis*-regulatory elements characterized by longer stretches of DNA, a higher level of histone H3K27 acetylation were defined as super-enhancers (SEs)[7,8]. They are bound with large amounts of transcription factors (TF) and coactivators to drive lineage-specific gene expression[9–11]. Malfunction of SEs leads to unleashed inflammation and subsequent autoimmune diseases such as atherosclerosis[12–14]. The mechanism by which SEs boost transcription remained unclear until recently, a liquid–liquid phase separation (LLPS) model was proposed[15]. LLPS forms non-membranous liquid compartments through weak, multivalent interactions among proteins and/or nucleic acids[16]. These liquid compartments serve as scaffolds to concentrate proteins that perform similar functions or a suppression model to make protein complexes unapproachable[17,18]. LLPS modulates various biological reactions, including chromatin structure remodeling[19,20], RNA processing[21], signal transduction[22–24], and SE regulated transcription[25–27]. Through LLPS, coactivators recruit TFs together with SEs to achieve a long-range interaction between distal regulatory elements and promoters and consequently promote transcription of multiple genes simultaneously with high efficiency[15]. It has been illuminated that various extracellular stimuli prime and activate the subset-specific SE profile in macrophages[28]. However, how those enhancers, especially latent SEs being silenced or fine-tuned to control the magnitude of the macrophage-mediated immune response, is still unclear.

The epigenetic reader protein ZMYND8 recognizes various histone codes, including H3K4 methylation and H3K14 acetylation[29,30]. In response to DNA breaks, it incorporates into a suppressive complex, facilitating the DNA damage repair[31,32]. ZMYND8 can deactivate enhancers through H3K4me3 demethylase KDM5C and repress oncogenes transcription and subsequent tumorigenesis[33]. Through collaboration with BRD4, HIF-1, and HIF-2, ZMYND8 promotes gene transcriptions in breast cancer cells[34]. This positive regulation effect may also be due to the association between ZMYND8 and P-TFEB complex[35].

Here we demonstrated that ZMYND8 forms phase-separated liquid compartments both in vitro and in vivo. Dependent on its liquidity feature, ZMYND8 compartments undergo dynamic translocation onto NF-κB associated latent SEs and suppress the subsequent pro-inflammatory genes expression through LSD1-mediated enhancer deactivation. We demonstrated LLPS could both stimulate and decommission SEs and propose a phase separation model of spatiotemporal transcription control. The epigenetic reader coalesces with post-translationally modified TFs to increase the regulatory specificity and efficiency on SEs-related gene transcription.

## Results

**ZMYND8 suppresses pro-inflammatory gene expression and alleviates the macrophage-mediated inflammatory response in vivo.** Given the reduced ZMYND8 expression in multiple inflammatory disease patients[36–39] (Supplementary Fig. 1a, b), we focus on the function of ZMYND8 in macrophages-mediated inflammation. As such, we generated *Zmynd8* conditional knockout mice (cKO) by crossing *Zmynd8*-floxed mice with *Lyz2*-Cre mice (Supplementary Fig. 1c, d). *Zmynd8* cKO mice are viable and have normal myeloid lineage cell development in bone marrow, including macrophages (Supplementary Fig. 1e), neutrophils (Supplementary Fig. 1f), and dendritic cells (Supplementary Fig. 1g). In addition, *Lyz2*-Cre-mediated *Zmynd8* deletion does not influence the in vitro differentiation of bone-marrow-derived macrophages (BMDMs) (Supplementary Fig. 1h).

We then in vitro induced classical (M1 type) and alternative (M2 type) macrophage polarization with LPS and IL-4, respectively. Polarized WT and *Zmynd8* cKO BMDMs were applied to transcriptome analysis by RNA-seq. Most differentially expressed genes are upregulated in *Zmynd8* cKO BMDMs after LPS treatment (Fig. 1a). Especially pro-inflammatory gene expression, such as *Ccl2*, *Il1b*, *Il6*, *Cxcl10*, and *Nos2* (Fig. 1b). Gene set enrichment analysis (GSEA) suggested that upregulated genes are enriched in the NF-κB pathway (Fig. 1c). In contrast, WT and *Zmynd8*-deficient BMDMs exhibited a similar transcriptome change in the non-treated (NT) and IL-4 treatment groups (Supplementary Fig. 2a, b). LPS-induced upregulation results were also confirmed by RT-qPCR (Supplementary Fig. 2c). Meanwhile, Pam3CSK4 stimulation also induces elevated pro-inflammatory gene expression in *Zmynd8* cKO BMDMs through the TLR2 pathway (Supplementary Fig. 2d). Marker genes in alternatively polarized macrophages (IL-4 treatment), such as *Arg1*, *Mrc1*, and *Pdl2*, did not differ significantly without *Zmynd8* (Supplementary Fig. 2e). We also generated the *Zmynd8*-deficient Raw264.7 mouse macrophage cell line (Supplementary Fig. 2f) and observed similar results, indicating that ZMYND8 suppresses pro-inflammatory gene expression (Supplementary Fig. 2g). The protein level of ZMYND8 is pretty much stable until 24 h post-LPS polarization (Supplementary Fig. 2h). *Zmynd8* loss did not alter the TLR4 signal transductions, as evidenced by the similar extent of phosphorylation levels of ERK and p38 (Supplementary Fig. 3a), as well as the JNK and NF-κB/ p65 in BMDMs (Supplementary Fig. 3b). Similar results were present in Raw264.7 cells (Supplementary Fig. 3c, d).

Dextran sodium sulfate (DSS) induces macrophage-mediated colitis in vivo independent of T or B cells[40]. We, therefore, set up an inflammatory bowel disease (IBD) model in *Zmynd8* cKO mice by treating them with DSS. In *Zmynd8* cKO mice, macrophages induced severe inflammation symptoms, reflected by quicker weight loss (Fig. 1d) and shorter colons (Fig. 1e, f) than WT controls. Histological investigations verified the severe inflammatory phenotypes, evidenced by thicker colonic walls and more severe mucosal structure disruption in *Zmynd8* cKO mice (Fig. 1g). Flow cytometry analysis of infiltrated immune cells in the colons from DSS-treated mice show comparable CD45+ cells and CD4+ and CD8+ T cells between WT and *Zmynd8* cKO mice. However, *Zmynd8* deficiency led to enhanced macrophage infiltration into colons (Fig. 1h). We also measured the cytokines in the sera from DSS-treated mice and found that both pro-inflammatory cytokines IL-6 and IL-1β were elevated in *Zmynd8* cKO mice (Fig. 1i). All those data demonstrated an exacerbated inflammatory response in DSS-treated *Zmynd8* cKO mice.

High-fat-diet (HFD)-treated mice also develop a macrophage-mediated, inflammation-dependent obesity phenotype[41]. We fed WT and *Zmynd8* cKO mice with an HFD for 16 weeks. Upon HFD challenge, *Zmynd8* cKO mice gained more body weight (Fig. 1j). They displayed a higher weight of multiple adipose depots (Fig. 1k), including gonadal white adipose tissue (gonWAT) and subcutaneous white adipose tissue (subWAT). Meanwhile, the glucose tolerance test (GTT) assay showed that *Zmynd8* cKO mice had a much slower glucose clearance rate

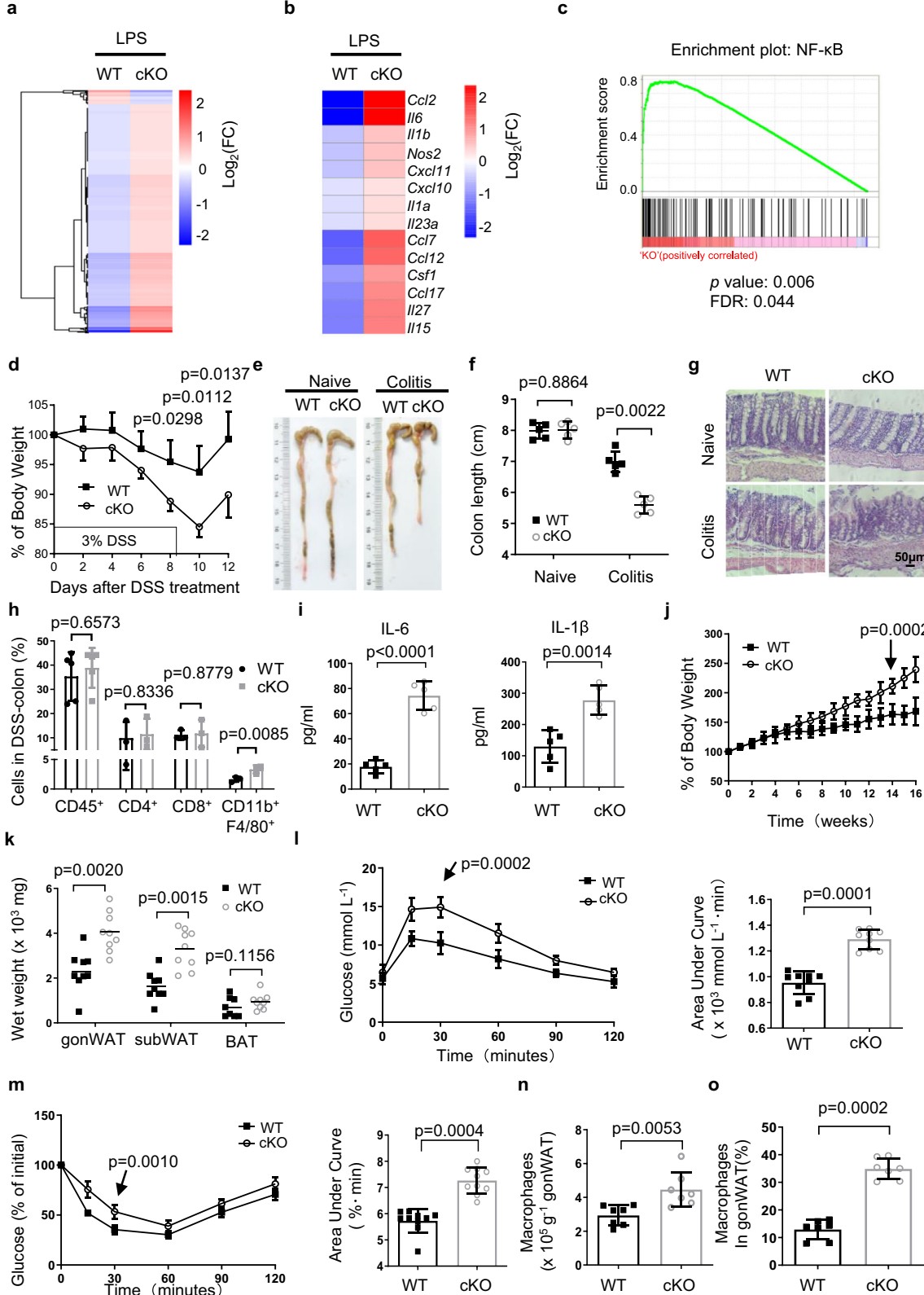

(Fig. 1l). The area under the curve (AUC) of GTT confirmed that the lack of *Zmynd8* in macrophages decelerates glucose tolerance (Fig. 1l). Insulin tolerance test (ITT) assay showed that blood glucose levels in *Zmynd8* cKO mice decreased more slowly than the WT control after insulin injection (Fig. 1m), indicating less sensitivity to insulin administration in *Zmynd8*-deficient macrophages. FACS measurements showed that the gonWAT tissue from obese *Zmynd8* cKO mice contained more macrophages in absolute numbers (Fig. 1n) and frequencies (Fig. 1o) compared to those in WT mice. Taken together, both models demonstrated that *Zmynd8* deficiency in macrophages aggravates the inflammatory response in vivo.

**Fig. 1 ZMYND8 suppresses pro-inflammatory gene expression and alleviates the macrophage-mediated inflammatory response in vivo. a, b** RNA-Seq analysis of WT and *Zmynd8* cKO BMDMs stimulated with LPS for 6 h. The heatmap analysis showing the differentially expressed genes with adjusted *p* value <0.05, false discovery rate <0.05, and log$_2$ fold-change >1.2 (**a**). The indicated pro-inflammatory cytokine and chemokine genes are upregulated in *Zmynd8* cKO BMDMs (**b**). Two-tailed Student's *t*-test determined *p* values. **c** GSEA analysis of RNA-Seq data obtained from WT and *Zmynd8* cKO BMDMs treated with LPS. NF-κB pathway gene cluster was enriched in *Zmynd8* cKO BMDMs. **d** Five 6-week-old female WT and *Zmynd8* cKO mice were challenged with DSS in drinking water for 7 days, and the body weight loss was measured from day 0. DSS-treated mice were euthanized on day 12. Two-tailed Student's *t*-test determined *p* values. **e, f** The macroscopic images of colons were shown (**e**), and colon lengths were measured (*n* = 5 per group). *P* values were determined by a two-tailed Student's *t*-test (**f**). **g** Representative H&E-stained images of proximal colon cross-section of naïve and DSS-challenged WT and *Zmynd8* cKO mice at day 12. Bars, 50 μm. **h** The percentage of the colon-infiltrated immune cells of DSS-challenged WT and *Zmynd8* cKO mice (*n* = 4 per group) was evaluated by flow cytometry on day 6. **i** ELISA measurement of pro-inflammatory cytokines production in the blood of WT and *Zmynd8* cKO mice (*n* = 5 per group) challenged with DSS for 7 days. **j** Body weight changes of WT and *Zmynd8* cKO mice (*n* = 10 per group) during HFD feeding. **k** Depot mass of gonWAT, subWAT, and BAT in WT and *Zmynd8* cKO mice (*n* = 9 per group). **l, m** GTT (*n* = 8 per group) (**l**) and ITT (*n* = 9 per group) (**m**) assays were performed in WT and *Zmynd8* cKO mice. **n, o** The absolute numbers (**n**) and frequencies (**o**) of gonWAT macrophage in HFD fed WT and *Zmynd8* cKO mice (*n* = 7 per group) based on flow cytometry analysis. All data are means ± SD. Two-tailed Student's *t*-test determined *p* values. All the animal experiments were repeated twice. Source data are provided as a Source Data file.

**LPS-induced ZMYND8 translocation onto the latent SEs regulating pro-inflammatory genes**. The genome-wide-binding patterns of ZMYND8 in non-treated (NT) and LPS-treated BMDMs were analyzed by ChIP-seq, respectively. As reported before[33], ZMYND8 bound promoter regions, but with a higher preference for intergenic and intron regions of the genome (Supplementary Fig. 4a, left). LPS stimulation did not change this binding preference (Supplementary Fig. 4a, right). Though the total numbers of ZMYND8-binding peaks decreased after LPS treatment, we observed 4393 peaks newly occupied by ZMYND8 (Fig. 2a), indicating that LPS induces genomic redistribution of ZMYND8. We then did a comprehensive ChIP-Seq analysis of enhancer markers, H3K4 mono-methylation (H3K4me1), H3K27 acetylation (H3K27ac), and chromatin accessibility analysis by ATAC-Seq in NT and LPS-treated BMDMs. ZMYND8 associated loci highly overlapped with H3K4me1 and H3K27ac peaks (Supplementary Fig. 4b), representing the active enhancer regions[42,43].

According to the ZMYND8 distributions in NT- and LPS-treated macrophages, we classified ZMYND8-binding peaks into three groups, LPS-gained, LPS-lost, and constitutively bound (Fig. 2a). We noticed that LPS-gained ZMYND8 peaks were highly enriched in the latent enhancer regions, which initially lacked H3K4me1 and H3K27ac markers and exhibited less chromatin accessibility, but underwent epigenetic remodeling to acquire active enhancer markers and chromatin accessibility after LPS treatment (Fig. 2b). However, ZMYND8 LPS-lost and constitutively bound peaks did not exhibit this preference (Supplementary Fig. 4c, d). After converting ZMYND8 chromatin-binding peaks into related genes, LPS-gained ZMYND8-binding peaks are associated with 1512 genes; LPS-lost ZMYND8 peaks are linked to 6891 genes, and 3501 genes bearing constitutively bound ZMYND8 peaks. Based on the RNA-Seq result, the expression of those LPS-gained ZMYND8 peaks related genes (LPS-gained genes) is significantly upregulated in *Zmynd8*-deficient BMDMs. In contrast, constitutively bound and LPS-lost genes were not dramatically changed (Fig. 2c).

We also analyzed the ZMYND8 distribution on the SEs by applying the ZMYND8-binding peaks to the Rank Ordering of Super-Enhancers (ROSE) algorithm and analyzed the ZMYND8 abundance at those SE regions. We identified a total of 409 SEs bound with ZMYND8 in NT BMDMs, and an increased number of 646 SEs in LPS-treated BMDMs (Fig. 2d). Some SEs, such as *Mir704* and *Ezh2*, were constitutively activated and showed comparable ZMYND8-binding intensities in both NT and LPS treated BMDMs. While at enhancer regions related to *Mir339*, *Lrrc28*, and *Lipa*, ZMYND8-binding peaks were diminished from

those sites. Interestingly, LPS-gained ZMYND8 peaks accumulated on the newly formed SEs, such as *Ccl2* and *Il1b* (Fig. 2d, right). Genome browser snapshots confirmed the translocation of LPS-gained ZMYND8 peaks onto latent SEs, including *Ccl2*, *Ccl5*, *Il1b*, and *Cxcl10*. These SEs are previously inaccessible and lack enhancer markers H3K4me1 and H3K27ac before TLR4 pathway activation (Fig. 2e). Thereby, ZMYND8 preferably moves onto latent SEs de novo generated in LPS polarized macrophages.

In total, 362 genes are associated with LPS-induced latent SEs, among which the 259 SE regulating genes are overlapped with genes related to LPS-gained ZMYND8 peaks (Fig. 2f). We found that those 259 genes were also mainly upregulated in *Zmynd8*-deficient BMDMs (Fig. 2g). KEGG (Kyoto Encyclopedia of Genes and Genomes) analysis on those 259 latent SE genes indicates multiple inflammatory pathways, including NF-κB, which are reminiscent of RNA-seq results (Fig. 2h). These results suggest that ZMYND8 undergoes genomic redistribution after classical polarization, and LPS-gained ZMYND8-binding peaks suppress the latent SEs of pro-inflammatory genes.

**ZMYND8 forms phase-separated liquid condensates in the nucleus**. LLPS highly controls the activation of SE to achieve long range and efficient regulations[25]. To evaluate whether ZMYND8 also employs the same mechanism in macrophages to undergo broad redistributions in response to a classical polarization signal, we first assessed the intracellular localization of endogenous ZMYND8 by immunofluorescence. ZMYND8 is mainly localized in the nucleus and forms spheric protein puncta or nuclear speckles with an average size of 100–300 nm (Fig. 3a). Then we overexpressed the GFP-tagged ZMYND8 protein in macrophage cell line Raw264.7 and measured the fluidity of ZMYND8 compartments by using fluorescence recovery after photobleaching (FRAP). The bleached ZMYND8 puncta recovered their GFP fluorescence quickly (Fig. 3b), verifying the fluidity and protein diffusion coefficients of the compartments formed by ZMYND8. The intrinsically disordered regions (IDR), or low complexity domains (LCD), embedded in proteins are responsible for LLPS[17]. We analyzed the ZMYND8 protein sequence by online analysis software, Predictor of Natural Disordered Region (PONDR) (v.VL3). We found that ZMYND8 has an uncharacterized protein domain (from amino acids 388 to 899) with the highest IDR predictive score value (Fig. 3c). We generated various ZMYND8 protein fragments to verify that the IDR domain of ZMYND8 executes LLPS (Fig. 3d). The nuclear-localized IDR domain forms liquid compartments directly in transfected HeLa cells, reflected by the FRAP (Fig. 3e and Supplementary Fig. 5a, b) and its ability to undergo droplet fusion (Fig. 3f). The deletion of the C-terminal MYND (Myeloid-Nervy-DEAF1) domain does

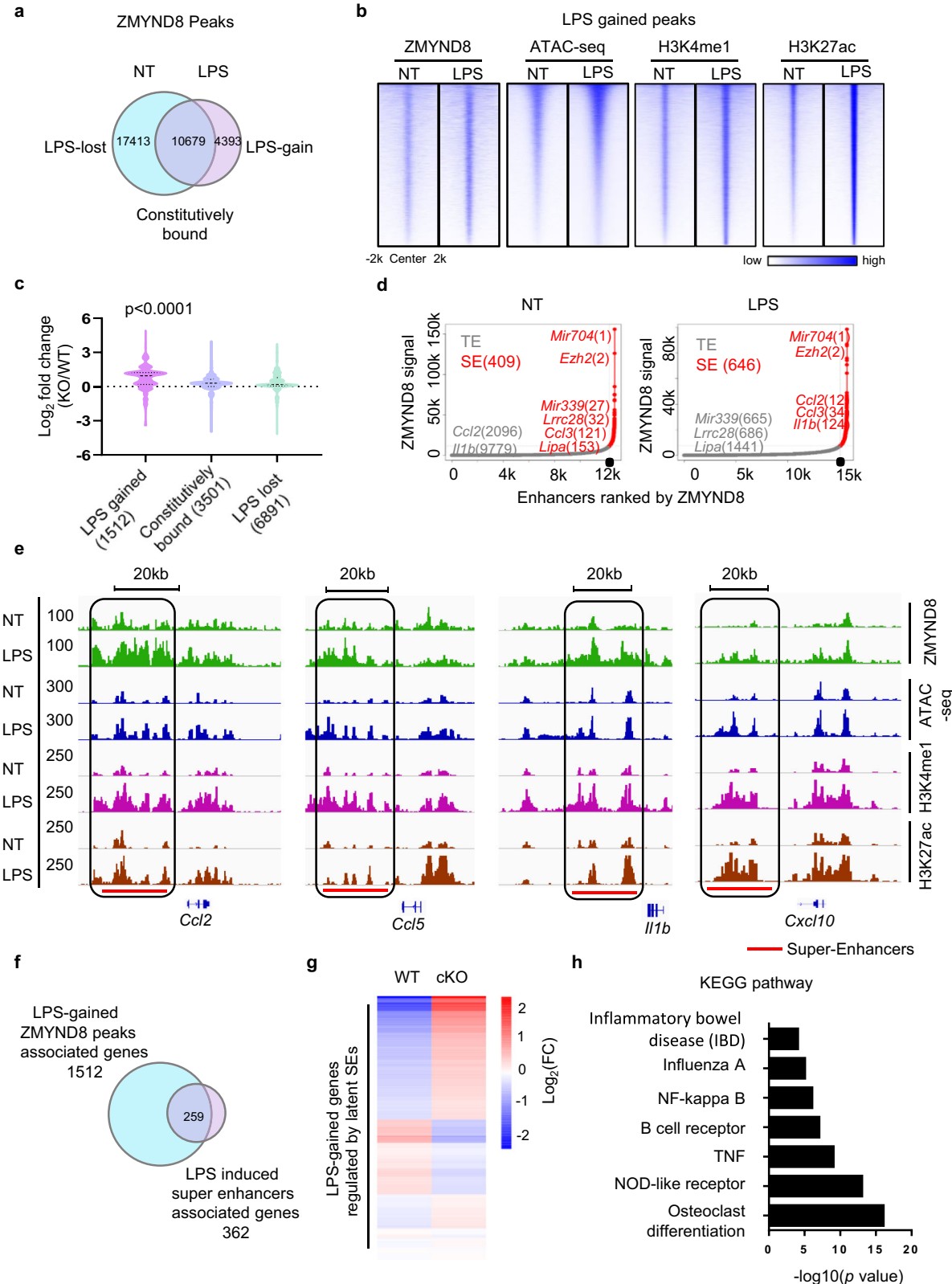

not affect the liquid compartment formation (Supplementary Fig. 5c). The N-terminal PBP domain (containing PHD, BRD, and PWWP domains only, which recognize multiple histone modification codes) cannot form the liquid compartments in transfected cells (Supplementary Fig. 5d). To confirm ZMYND8 itself can form those liquid compartments, we then expressed and purified the GFP-tagged ZMYND8-IDR domain from *E. coli*. In vitro-purified IDR protein forms liquid-like condensates in a protein concentration-dependent manner (Fig. 3g). 1,6-Hexanediol (1,6-HD) is an aliphatic alcohol, which would not change protein stability but can disassemble phase-separated condensates by disrupting the weak hydrophobic interactions[44]. ZMYND8-

**Fig. 2 LPS-induced ZMYND8 translocation onto the latent SEs regulating pro-inflammatory genes. a** Venn diagram analysis of ChIP-seq peaks of ZMYND8 in non-treated (NT) and LPS-treated BMDMs. LPS-gained, LPS-lost, and constitutively bound ZMYND8 peaks were labeled. **b** Heatmap analysis of ChIP-seq data obtained from NT and LPS-treated BMDMs. Peak density heatmap showing LPS-gained ZMYND8-binding signals, where the H3K4me1, H3K27ac, and chromatin accessibility (ATAC-seq) signals before and after LPS treatment were also shown. Each row shows ±2 kb centered regions of LPS-gained ZMYND8 peaks. **c** Based on the RNA-Seq results from WT and *Zmynd8* cKO BMDMs, the violin plots summarizing the *Zmynd8* deficiency-induced expression change of genes associated with LPS-gained, LPS-lost, and constitutively bound ZMYND8 peaks. *P* value by one-way ANOVA test. **d** Ranked plots of enhancers and super-enhancers defined in NT (left) and LPS-treated (right) BMDMs associated with increasing ZMYND8 signals (units: r.p.m.). Enhancers and SEs are defined as regions of ZMYND8 ChIP-seq binding not contained in promoters. The cutoff discriminating TEs from SEs is shown as a dashed line. Genes associated with enhancers that are considered typical or super are colored gray and red, respectively. **e** Genome Browser tracks showing ZMYND8, H3K4me1, H3K27ac ChIP-seq signals, and chromatin accessibility (ATAC-Seq) in the NT and LPS-treated macrophages at selected SE regions. **f** Genes associated with LPS-gained ZMYND8 peaks are overlapped with those LPS-induced latent SEs. **g** Heatmap analysis showing *Zmynd8* deficiency-induced expression changes of genes regulated by ZMYND8-enriched latent SEs. **h** KEGG pathway analysis of genes associated with both LPS-gained ZMYND8 peaks and LPS-gained SEs. Source data are provided as a Source Data file.

IDR liquid compartments are sensitive to the 1,6-HD treatment (Fig. 3h). Meanwhile, the high sodium chloride concentration also disperses the ZMYND8-IDR liquid droplets in vitro (Fig. 3i).

LLPS relies on the low affinity, multivalent interactions mediated by the charged amino acids inside IDR regions[16]. We found the ZMYND8-IDR region has a large portion of negatively charged amino acids D/E (aspartic acid/glutamic acid) (Supplementary Fig. 5e). We mutated all those 66 acidic residues on the IDR region into Alanine (A) and investigated the effects on LLPS. Mutated ZMYND8 in full-length (FL) still forms spheric puncta in the nucleus, but those compartments lost their liquidity and failed to recover GFP fluorescence in the FRAP assay (Fig. 3j). Purified WT IDR fragment (Supplementary Fig. 5f) fully recovered in the FRAP assay in vitro (Fig. 3k, red line). In comparison, the protein condensates formed by ZMYND8 D/E-to-A mutant (Supplementary Fig. 5g) failed to recover the GFP fluorescence in the FRAP assay (Fig. 3k, blue line). Therefore, D/E-to-A mutations on the ZMYND8-IDR region intensify the protein–protein interactions, consolidating the ZMYND8 liquid droplets into solid aggregates.

All those data suggested that ZMYND8 can perform LLPS both in vivo and in vitro, and the acidic amino acids in the IDR region of ZMYND8 play indispensable roles in the LLPS.

**ZMYND8 translocation onto SEs is dependent on its liquidity feature**. To investigate whether ZMYND8 liquid compartments are present directly at the SEs of pro-inflammatory genes, we designed and synthesized multiple DNA probes targeting the *Ccl2* SE and applied DNA fluorescence in situ hybridization (FISH) after immuno-staining the endogenous ZMYND8. In LPS-treated macrophages, ZMYND8 speckles are colocalized with the SE region upstream of the *Ccl2* gene, which indicates that ZMYND8 occupied those SEs through liquid condensates (Fig. 4a). In vitro, 1,6-HD dissolved the FL ZMYND8 liquid droplets efficiently (Supplementary Fig. 5h). We then applied 1,6-HD onto LPS-treated BMDMs to evaluate the ZMYND8 association with SEs by ChIP-seq and FISH. 1,6-HD treatment did not affect the transcription or the ZMYND8 protein stability (Supplementary Fig. 5i, j) but disrupts the ZMYND8 binding of total peaks from ChIP, especially the LPS-gained peaks (Fig. 4b). ZMYND8-binding peaks at the SEs of *Ccl2*, *Ccl5*, *Il1b*, and *Cxcl10* were also diminished by 1,6-HD (Fig. 4c). The disassociation of ZMYND8 from the SE region after 1,6-HD treatment was further supported by the FISH results (Supplementary Fig. 5k).

To further confirm that LLPS mediated by ZMYND8 is critical for suppressing the SEs and related pro-inflammatory genes, we introduced the ectopic expression of WT *Zmynd8* and D/E-to-A mutant into *Zmynd8* deficient Raw264.7 cells. D/E-to-A mutant lost the liquidity characteristic, although it still forms protein puncta in the nucleus. When focusing on the regions where LPS-

gained ZMYND8 peaks occupied (Fig. 2a), the intensity of the D/E-to-A mutant-binding peaks in non-treated (NT) cells was comparable to the WT ZMYND8 peaks. But after LPS stimulation, WT ZMYND8 gained much stronger signals when compared to D/E-to-A mutant-binding peaks (Fig. 4d). Genome browser snapshots confirmed that, before LPS treatment, both WT and D/E-to-A mutant have a similar binding pattern and intensity on those SEs (Fig. 4e, upper), which indicated a comparable genome binding ability between WT and ZMYND8 mutant. LPS treatment causes elevated WT ZMYND8-binding peaks on the SEs regions of pro-inflammatory genes, such as *Ccl2*, *Ccl5*, *Il1b*, and *Cxcl10*, but the D/E-to-A mutant-binding intensity was not changed (Fig. 4e, bottom). Meanwhile, according to the RNA-Seq results from WT ZMYND8 and D/E-to-A mutant Raw264.7 cells after LPS treatment, the majority of the 259 latent SEs regulated genes, which are also occupied by LPS-gained ZMYND8 peaks, were elevated in the D/E-to-A mutant cells (Fig. 4f). These SEs' genes upregulated in D/E-to-A cells mainly overlap with the latent SE genes upregulated in *Zmynd8* cKO cells (Fig. 4g).

These data emphasized that the formation of ZMYND8 liquid condensates is critical for the translocation of ZMYND8 onto latent SEs in response to the LPS stimulation. The aggregation of the ZMYND8 D/E-to-A mutant represents a KO phenotype.

**p65 guides the redistribution of ZMYND8 liquid condensates onto NF-κB-associated SEs**. Both the GSEA analysis of RNA-Seq data and the KEGG analysis of the ChIP-Seq data indicate ZMYND8 target genes highly correlated with the NF-κB pathway, so we hypothesized that the dynamic ZMYND8 translocation is coincident with NF-κB TFs after LPS stimulation. Protein over-expression in HEK-293 T cells suggested the interaction between ZMYND8 and p65 (Rel A) (Fig. 5a), and the endogenous co-immunoprecipitation results also confirm ZMYND8 associates with p65 in WT BMDMs only after LPS stimulation (Fig. 5b). Immunofluorescence assay shows that endogenous ZMYND8 co-localizes with p65 after LPS stimulation (Fig. 5c). But in untreated BMDMs, ZMYND8 and p65 immunofluorescence signals were rarely overlapped (Fig. 5d and Supplementary Fig. 6a). To evaluate the ZMYND8 and p65 colocalization is in an LLPS-dependent manner, we further incubated purified GFP-tagged ZMYND8-IDR protein together with Cherry-tagged p65 in phase separation buffer in vitro. p65 alone forms droplets in the LLPS buffer (Supplementary Fig. 6b). Several other transcriptional factors, such as MYC and p53, were included as specificity control (Supplementary Fig. 6c–f). Judged by a larger size of liquid compartments, ZMYND8 shows the best coalescent efficiency and fluidity with p65 compared to other TFs, like MYC and p53 (Fig. 5e, f). Notably, the ZMYND8 D/E-to-A mutant does not bind p65 in the protein immunoprecipitation assay (Fig. 5g). We

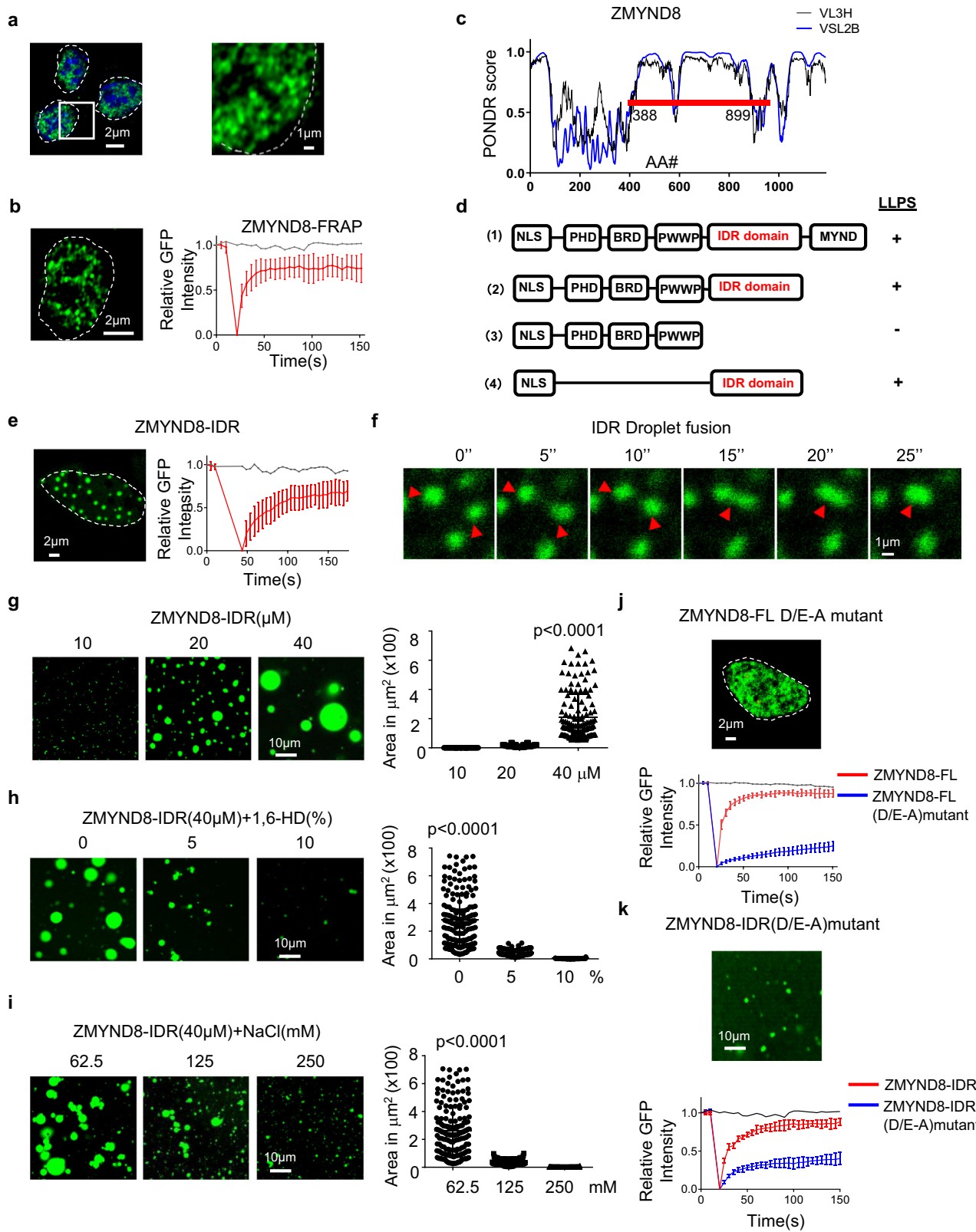

further evaluated the coalescent efficiency of p65 with WT IDR and IDR D/E-to-A mutant in vitro. IDR D/E-to-A mutant was incompetent to coalesce with p65 droplets (Fig. 5h), and thereby p65 alone forms much smaller droplets than p65 coalescent with WT IDR (Fig. 5i). In Raw264.7 cells, immunofluorescence data suggested that the D/E-to-A mutant also failed to interact with endogenous p65 (Fig. 5j). These data indicate that, in addition to

the epigenetic reader domain, the IDR domain of ZMYND8 is essential to the ZMYND8 dynamic translocation onto NF-κB-associated SEs.

Finally, we pursued p65 ChIP-Seq analysis in classically polarized macrophages. Around 50% of ZMYND8 peaks and 80% of ZMYND8-binding genes are co-occupied with p65 after LPS stimulation (Supplementary Fig. 6g). The metagene analysis

**Fig. 3 ZMYND8 forms liquid-liquid phase separation in the nucleus. a** Immunofluorescence imaging of endogenous ZMYND8 in BMDMs. ZMYND8 signal (green) is shown merged with DAPI stain. Experiments were repeated three times. **b** FRAP experiment of GFP-ZMYND8 overexpressed in Raw264.7 cells (left), quantification of FRAP data for GFP-ZMYND8 puncta (right). Bleaching event occurs at $t = 20$ s. Data are plotted as mean values ± SD ($n = 10$). **c** Graphs plotting intrinsic disorder regions (PONDR VSL2B and VL3H) of ZMYND8. PONDR VSL2B and VL3H score ($y$-axis) and amino acid position ($x$-axis) are shown. The red bar designates the IDR under investigation. **d** Schematic of recombinant GFP fusion ZMYND8 fragments tested for LLPS. **e** ZMYND8-IDR fragment was transfected into HeLa cells and analyzed by FRAP (left), quantification of FRAP data for ZMYND8-IDR fragment puncta (right). Data are plotted as mean values ± SD ($n = 10$, examined over three independent experiments). **f** Droplet fusion is highlighted in a higher resolution and extended times frame as indicated. **g** Representative images of GFP-ZMYND8-IDR droplet formation at different protein concentrations. Data are presented as mean values ± SD (10 µM: $n = 107$; 20 µM: $n = 116$; 40 µM: $n = 132$; examined over three independent experiments). $P$ value by one-way ANOVA test. **h** 1,6-Hexanediol (HD) was added at the indicated concentrations into solutions containing 40 µM GFP-ZMYND8-IDR. Data are presented as mean values ± SD (0%: $n = 169$; 5%: $n = 160$; 10%: $n = 166$; examined over three independent experiments). $P$ value by one-way ANOVA test. **i** Representative images of droplet formation at different salt concentrations. Forty micromolar GFP-ZMYND8-IDR was added to the droplet formation buffer. Data are presented as mean values ± SD (62.5 mM: $n = 169$; 125 mM: $n = 160$; 250 mM: $n = 166$; examined over three independent experiments). $P$ value by one-way ANOVA test. **j** All aspartic acid (D) and glutamic acid (E) residues in IDR were mutated to alanine (A) to generate mutant D/E-to-A (D/E-A). Then full-length (FL) WT ($n = 7$ samples), ZMYND8 (red line), and mutant ($n = 10$ samples) (blue line) were transfected into HeLa cells (upper), followed by FRAP quantification assay (lower). Data are presented as mean values ± SD. Cells were examined over three independent experiments. **k** Representative images of droplet formation of GFP-ZMYND8-IDR (D/E-A mutant) in vitro (upper) and analyzed by FRAP (lower). Red line: WT IDR ($n = 10$ samples); blue line: D/E-A mutant ($n = 9$ samples). Data are presented as mean. values ± SD. Source data are provided as a Source Data file.

---

of LPS-gained ZMYND8 peaks revealed the strong colocalization between p65 and ZMYND8, which is absent in the LPS-lost peaks (Fig. 5k). LPS-induced ZMYND8-binding peaks at SEs of pro-inflammatory genes are colocalized with p65 peaks after LPS stimulation (Fig. 5l), suggesting the redistribution of ZMYND8 onto specific SEs is likely dependent on NF-κB/p65.

**LPS-induced p65 acetylation at the K122 residue directs the docking of ZMYND8 liquid condensates onto NF-κB associated SEs.** Post-translational modifications on NF-κB, such as p65 acetylation, play critical roles in NF-κB activation. LPS stimulation induces multiple site-specific acetylations, which might improve the interaction between p65 and ZMYND8. As such, we mutated seven potential acetylated lysines on p65, K122, K123, K218, K221, K310, K314, and K315 into Arginine and investigated the site-specific influence on their interactions with ZMYND8. We identified that the K122R mutant abrogated the interaction between p65 and ZMYND8 (Fig. 6a). Interestingly, K122 acetylation on p65 was reported as a suppressive modification, but the detailed mechanism remains unknown[45]. We confirmed the endogenous K122 acetylation by immunoblot and found that K122 acetylation was only present in the nuclear fraction of p65 after LPS stimulation (Fig. 6b). Next, we performed in vitro acetylation on purified recombinant WT p65 and K122R mutant (Supplementary Fig. 7a), followed by incubating with ZMYND8-IDR protein in the in vitro LLPS assay. Compared to unmodified p65, acetylated WT p65 (Supplementary Fig. 7b, c) has much better coalescent efficiency and forms larger liquid droplets with ZMYND8. While in vitro acetylation on K122R mutant has little influence on the coalescence with ZMYND8 (Fig. 6c, d). We further applied in vitro His-pull-down assay to evaluate the acetylation-dependent p65 interaction with ZMYND8. The interaction between FL ZMYND8 and acetylated WT p65 was stronger than un-acetylated WT p65. In contrast, the K122R mutant showed equal binding with FL-ZMYND8 before and after acetylation (Supplementary Fig. 7d). These data suggest that the K122-acetylated p65 further improves its interaction and fusion with ZMYND8 liquid droplets.

We also investigated the protein domain of ZMYND8 responsible for the interaction with acetylated p65. We found the deletion of the IDR domain or MYND domain of ZMYND8 disrupted the interaction with p65. In contrast, the deletion of the BRD domain or acetyl-binding sites mutations (Y247A/N248A)[29] did not influence the interaction between p65 and ZMYND8

(Supplementary Fig. 7e). These results indicate that the ZMYND8 recognizes the acetylated p65 through the IDR instead of the BRD domain. However, as a histone acetylation reader, the BRD domain is still critical for the suppressive effect of ZMYND8 on gene expression. When we transduced Y247A/N248A (AA) mutated ZMYND8 into KO Rwa264.7 cells, we found that AA mutant failed to suppress the pro-inflammatory gene expression compared to WT ZMYND8 (Supplementary Fig. 7f). The AA mutant showed less binding on the SE regions of those pro-inflammatory genes (Supplementary Fig. 7g). Thereby, p65 acetylation strengthens the association between ZMYND8 and p65, but ZMYND8 recruitment to the chromatin through its BRD domain is a prerequisite.

To evaluate the physiological function of the p65 K122R mutant, we generated p65-deficient Raw264.7 cells and then reintroduced the WT p65 and K122R mutant. We applied ZMYND8 ChIP-seq analysis to those genetically modified cells. The total ZMYND8 peaks in WT p65 and K122R mutant cells are comparable (Fig. 6e, left and Supplementary Fig. 7h), but the LPS-gained ZMYND8 bound peaks at enhancer regions were significantly reduced in K122R mutant cells (Fig. 6e, right and Supplementary Fig. 7i). Genome browser snapshots showed that ZMYND8 occupancy at SEs of several pro-inflammatory genes, such as *Ccl2*, *Ccl5*, *Il1b*, and *Cxcl10*, is attenuated in p65-deficient cells, which can be replenished by WT but not K122R p65 (Fig. 6f). It suggests that the translocation of ZMYND8 on the genome is dependent on the acetylation of the K122 residue on p65. ZMYND8 ChIP-qPCR assay confirmed the sequencing results (Supplementary Fig. 7j). Meanwhile, according to the RNA-Seq results from WT p65 and K122R mutant Raw264.7 cells after LPS treatment, most of the latent SEs regulated genes were elevated in the K122R mutant cells (Fig. 6g). The latent SEs regulated genes with enhanced transcription in K122R cells are mainly overlapped with latent SE genes upregulated in D/E-to-A cells (Fig. 6h). Therefore, in response to LPS stimulation, K122-acetylated p65 directs the re-location of ZMYND8 liquid condensates onto the NF-κB-associated latent SEs and represses the expression of the related genes.

**ZMYND8 liquid condensates activate LSD1 to decommission SEs.** ZMYND8 is an epigenetic reader, which lacks the functional domain for epigenetic remodeling. To clarify the suppressive cofactors in ZMYND8 liquid condensates, we applied gel filtration chromatography analysis using nuclear extracts from LPS

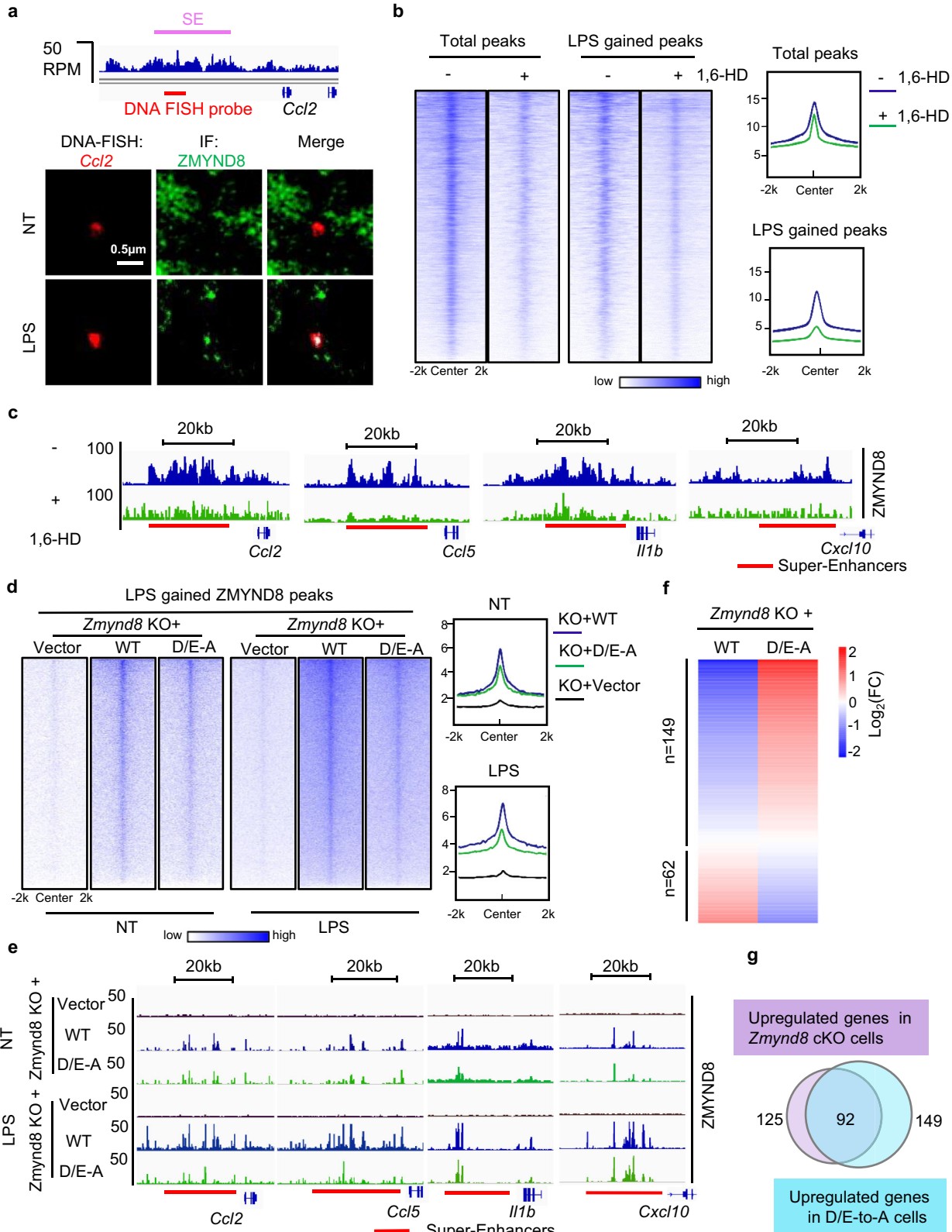

polarized control (EV) and *Zmynd8* KO Raw264.7 cells (Supplementary Fig. 8a). ZMYND8 is mainly present in the fractions containing the majority of the p65 (Fig. 7a). Meanwhile, histone modification enzymes, like LSD1, KDM5C, and NuRD complex component HDAC1, also accumulated in the same fractions (Fig. 7a). However, in *Zmynd8* KO cells, p65 is not enriched in those fractions and dispersed evenly into each fraction. At the

same time, LSD1 translocated from the high-molecular factions where ZMYND8 initially presents to the low-molecular fractions (Fig. 7b and Supplementary Fig. 8b). The distribution of KDM5C and HDAC1 was not altered by *Zmynd8* loss (Fig. 7b). Immunoprecipitation results showed that endogenous ZMYND8 could pull down LSD1 in the presence of p65 but not in p65-deficient Raw264.7 cells (Fig. 7c). ZMYND8 D/E-A mutant, which cannot

**Fig. 4 The translocation of ZMYND8 compartments onto SEs of pro-inflammatory genes is dependent on its liquidity feature. a** Depiction of *Ccl2* SE locus with ZMYND8 ChIP-seq signals (blue histograms) and the location of FISH probes. Colocalization between ZMYND8 protein and the *Ccl2* probe in fixed WT BMDMs by IF followed by DNA FISH. Images of the indicated IF and FISH are shown, along with the merged channels (overlapping signal in white). **b** Left, heatmap analysis showing the total and LPS-gained ZMYND8 ChIP-seq signals from the untreated (−) or 1,6-HD treated (+) BMDMs. Right, averaged profiles of the ZMYND8 ChIP-seq signals. All ChIP-seq signals are displayed from −2 to +2 kb surrounding the center of each annotated ZMYND8 peak. **c** Genome browser snapshot of ChIP-Seq peaks for ZMYND8 at indicated SE regions without (−) or with (+) 1,6-HD treatment in BMDMs. **d** Heatmap analysis showing ZMYND8 ChIP signals at the LPS-gained ZMYND8 peaks region in different types of Raw264.7 cells, including *Zmynd8* KO rescued with Vector (*Zmynd8* KO + Vector), *Zmynd8* KO with WT *Zmynd8* (*Zmynd8* KO + WT *Zmynd8*), and *Zmynd8* KO with D/E-A mutant (*Zmynd8* KO + D/E-A mutant), before (NT) and after LPS treatment (LPS). Averaged profiles of the ZMYND8 ChIP-seq signals are shown in the right panel. All ChIP-seq signals are displayed from −2 to +2 kb surrounding the center of each annotated ZMYND8 peak. **e** Genome browser view of ChIP-seq peaks for ZMYND8 at indicated SE regions in different types of Raw264.7 cells, including *Zmynd8* KO + Vector, *Zmynd8* KO + WT *Zmynd8*, and *Zmynd8* KO + D/E-A mutant. **f** Heatmap analysis of RNA-Seq results represent differentially expressed genes regulated by ZMYND8 enriched, latent SEs (from Fig. 2d) in *Zmynd8* KO + WT *Zmynd8* and *Zmynd8* KO + D/E-A mutant Raw264.7 cells. **g** The latent SEs' genes upregulated in D/E-to-A cells are overlapped with latent SE genes upregulated in *Zmynd8* cKO cells. Source data are provided as a Source Data file.

form liquid compartments either in vitro or in vivo, also failed to recruit LSD1 in Raw264.7 cells (Fig. 7d), indicating that the LLPS is critical for establishing the ZMYND8/p65/LSD1 complex. We found purified LSD1 proteins formed phase-separated liquid compartments directly in the phase separation buffer in vitro (Fig. 7e). But LSD1 failed to fuse with ZMYND8-IDR in the absence of p65 (Fig. 7f). When LSD1, p65, and ZMYND8-IDR proteins were incubated together in vitro, all three proteins fused into a coalescent liquid compartment (Fig. 7g). When LSD1 coalesced with liquid droplets formed by ZMYND8 and p65 in vitro, the fused complex appeared to be smaller with statistical significance (Supplementary Fig. 8c). We evaluated the liquid property of ZMYND8/p65/LSD1 compartments by FRAP assay. LSD1 decreases the FRAP recovery efficiency (Supplementary Fig. 8d), which indicates that LSD1 decreases the fluidity of the protein complex. These results suggest that besides p65, LSD1 is another critical functional component in the ZMYND8 liquid compartments. ZMYND8 recruits LSD1 in a p65-dependent manner.

Then we assessed the genome-wide-binding pattern of LSD1 in the WT and *Zmynd8* cKO BMDMs. The overall LSD1-binding preference in WT and *Zmynd8* cKO BMDMs did not differ significantly. However, when focusing on the LPS-gained ZMYND8 peaks region for analysis, LSD1-binding intensities were markedly reduced in *Zmynd8* cKO BMDMs (Fig. 7h). Genome Browser snapshot confirmed the LSD1 enrichment at the latent SEs regions (Fig. 7i). LSD1 is an H3K4 di- or mono-methylation demethylase, which suppresses gene transcription by silencing enhancers[45]. We thereby evaluate the H3K4 methylation status in BMDMs. WT and cKO BMDMs showed comparable H3K4me1 and H3K4me2 levels before LPS treatment (NT) (Supplementary Fig. 8e) and after LPS treatment as well (Supplementary Fig. 8f). Although ChIP-seq analysis results indicated that the overall H3K4me1 pattern in WT and *Zmynd8*-deficient BMDMs was not changed (Fig. 7j, left), H3K4me1 peak intensity at the LPS-gained ZMYND8 region was increased in *Zmynd8* cKO BMDMs (Fig. 7j, right). Genome Browser snapshot demonstrated that H3K4me1 modification at the latent SE regions of pro-inflammatory genes was enhanced in *Zmynd8* cKO BMDMs (Fig. 7k), and the increased H3K4me1 modification in *Zmynd8* cKO BMDMs is retained after LPS was eliminated (Supplementary Fig. 8g), which is a typical feature of latent enhancer[3,4]. The same upregulation trend was observed for the H3K4me2 peaks at the LPS-gained ZMYND8 region in *Zmynd8* cKO BMDMs (Supplementary Fig. 8h–j). Furthermore, the LSD1 ChIP-qPCR results demonstrated that LSD1 binding on the SEs of pro-inflammatory genes, such as *Ccl2*, *Il1b*, and *Il1a*, was attenuated in the ZMYND8 D/E-A mutant cells compared to that in WT cells (Supplementary Fig. 8k). Meanwhile, the H3K4me1

and H3K4me2 in those SE regions were upregulated (Supplementary Fig. 8l, m)

To examine the activity of enhancers, we examined the enhancer RNA (eRNA) transcription from several ZMYND8 targeted latent SEs, such as *Ccl2*, *Ccl5*, *Il1b*, and *Cxcl10*. The eRNA transcription level was dramatically increased in *Zmynd8*-deficient BMDMs (Fig. 7l), which indicates that ZMYND8 suppresses the expression of pro-inflammatory genes through decommissioning enhancers. Knock-down endogenous *Lsd1* by shRNA can dramatically elevate LPS-induced pro-inflammatory gene expression in Raw264.7 cells (Supplementary Fig. 8n), which is consistent with the effects caused by *Zmynd8* deficiency. Similarly, treatment with LSD1-specific inhibitors ORY-1001 led to the increased transcription of pro-inflammatory genes in WT BMDMs but did not further enhance the expression of the pro-inflammatory genes in *Zmynd8* cKO BMDMs (Supplementary Fig. 8o). These results support that ZMYND8 decommissioning enhancers, especially SEs, are mediated by LSD1.

**Discussion**

Our results suggest that ZMYND8 is a vital gatekeeper for macrophage-mediated inflammation. The in vivo relevance of our findings was evident in mouse models of colitis and obesity, which resulted in the aggravated disease upon conditional abla-tion of *Zmynd8* in macrophages. ZMYND8 can decommission a specific set of SEs through LLPS, and aberrant activation of those SEs determines the in vitro and in vivo phenotypes we observed. Although the establishment of a cell lineage-specific enhancer profile has been investigated[46], how enhancers, especially SEs, are tuned down or silenced is not well known. We found ZMYND8 undergoes genomic redistributions onto latent SEs and suppresses their activities through LSD1 catalyzing H3K4 demethylation. ZMYND8 recognizes histone codes, such as H3K4 methylation and H3K14 acetylation[29,30,33], which means the localization of ZMYND8 on the genome is less likely DNA sequence-specific. While NF-κB/p65 guides and assembles phase-separated ZMYND8 decommissioning complex onto specific SEs, which were de novo generated by p65 itself (Fig. 2d–f). NF-κB/p65 is the most critical SDTF in classically polarized macrophages[2], acti-vating latent enhancers to induce pro-inflammatory the robust expression of pro-inflammatory genes[28]. Thereby, our results indicate that p65 not only activates gene transcription but also helps decommission enhancers. The histone modifications on latent enhancers are retained as a memory for innate immunity, which will trigger a more rapid immune response for the re-stimulation[3,4]. By targeting NF-κB/p65-associated latent enhan-cers, ZMYND8 liquid compartments set up a threshold for the inflammatory reaction at the initial stage.

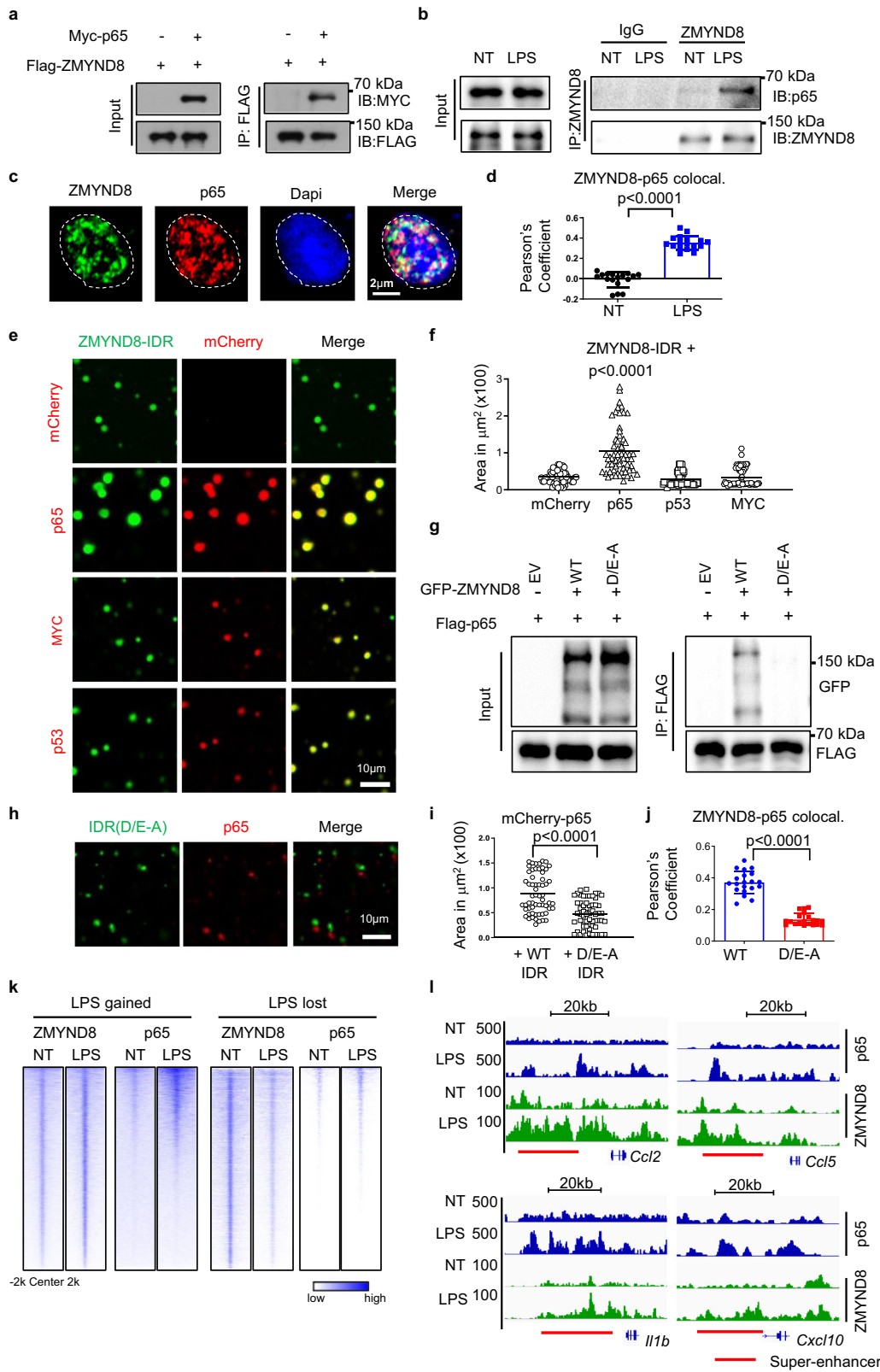

Our data also suggested that ZMYND8 decommissions specific SEs in a spatiotemporal manner. LPS-induced K122 acetylation on p65 increases the coalescent efficiency between p65 and ZMYND8 liquid compartments and then promotes ZMYND8 to execute a more specific and efficient regulation on latent SEs. Since K122 acetylation on p65 occurs 15 min after LPS treatment

(Fig. 6b), it provides a temporal regulation axis for ZMYND8-mediated enhancers deactivation, besides the spatial localization. The classical lock-and-key protein–protein interaction models are not preferred in the LLPS model when cargo proteins are recruited. Instead, it is demonstrated that a limited number of phase-separated transcriptional coactivators interact with many

**Fig. 5 p65 guides the redistribution of ZMYND8 liquid condensates onto NF-κB associated SEs. a** 293 T cells were transfected with Myc-p65 and Flag-ZMYND8 for 24 h, followed by Co-IP and western blotting with FLAG antibody. Experiments were repeated three times. **b** Endogenous Co-IP and western blotting with anti-ZMYND8 antibody in NT and LPS-treated BMDMs. Experiments were repeated three times. **c** Immunofluorescence staining of endogenous ZMYND8 (green) in primary mouse BMDMs shows colocalization with endogenous p65 (red) after treatment with LPS for 1 h, DAPI nuclear staining in blue. **d** Quantization of colocalization: the graph represents Pearson's coefficient of ZMYND8 and p65 from three independent treatments of primary mouse BMDMs. Colocalization of ZMYND8 to p65 shows a significant difference between non-treated (NT) and LPS-treated (LPS) cells. Data are presented as mean values ± SD ($n = 17$ cells). Two-tailed Student's $t$-test determined $p$ values. **e** Droplet formation of various TFs with GFP-ZMYND8-IDR in droplet formation buffers containing 125 mM NaCl and 10% PEG8000 in vitro. Ten micromolar of GFP-ZMYND8-IDR was added into 10 μM of mCherry, mCherry-p65, mCherry-MYC, and mCherry-p53, respectively. Representative images were shown. **f** Statistical analysis of the coalescent droplet size when GFP-ZMYND8-IDR mixed with mCherry, mCherry-p65, mCherry-p53, and mCherry-Myc, respectively ($n = 60$, examined over three independent experiments). P value by one-way ANOVA test. **g** 293 T cells were transfected with Flag-p65 and EV, GFP-ZMYND8, GFP-ZMYND8 mutant (D/E-A) for 24 h followed by Co-IP and western blot with anti-FLAG antibody. Experiments were repeated three times. **h** Representative images of separated droplet formation when mCherry-p65 was incubated with GFP-ZMYND8-IDR mutant (D/E-A). **i** Statistical analysis of the mCherry-p65 droplet size, when WT-ZMYND8-IDR and GFP-ZMYND8-IDR D/E-A mutant was added in droplet formation buffer respectively ($n = 60$, examined over three independent experiments). Two-tailed Student's $t$-test determined $p$ values. **j** The colocalization of p65 with WT ZMYND8 or D/E-A mutant in Raw264.7 cells was evaluated by IF. Data are presented as mean values ± SD ($n = 20$ cells, examined over three independent experiments). Two-tailed Student's $t$-test determined $p$ values. **k** Heatmap analysis of the colocalization of p65 ChIP-Seq signals with LPS-gained and LPS-lost ZMYND8 peaks. **l** Genome browser view of ChIP-seq peaks of ZMYND8 and p65 at indicated SE regions in NT and LPS treated BMDMs. Source data are provided as a Source Data file.

different TFs, indicating less specificity in this phase separation model[15,47]. Given that there is always a distinct transcription profile in the differentiated cell, our results proposed a mechanism by which post-translational modification facilitates the phase-separated regulators to select specific TFs and SEs.

1,6-HD treatment was widely employed to dissolve the phase-separated liquid compartments, and it did induce ZMYND8 dissociation from the chromatin. However, 1,6-HD treatment cannot distinguish the difference between the liquid compartment and the gel-like, solid protein aggregate, both present as a protein complex in the cells. To further evaluate the function of liquid compartments, a specific ZMYND8 mutant was involved in our functional assay. D/E-to-A mutations on the IDR region consolidate the ZMYND8 compartments into solid, irreversible protein aggregates. These solidified protein aggregates lost fluidity and failed to interact with p65 (Figs. 3k and 5g). The dynamic translocation of ZMYND8 mutant onto latent SEs, as well as subsequent suppressive effect, were both disabled (Fig. 4d–f). It also coordinates that only LPS-gained ZMYND8 peaks in LPS polarized macrophages have functional significance (Fig. 2c). Those data together strengthen that the LLPS of ZMYND8 plays an indispensable role in its spatiotemporal redistributions onto the latent SEs, which themselves undergo dynamic remodeling as well. When ZMYND8 compartments lose the liquidity feature, although protein expression level is normal, a *Zmynd8*-deficient phenotype is present.

Moreover, phase-separated ZMYND8 recruits the demethylase LSD1 in a p65-dependent manner to decommission latent enhancers and SEs. LSD1 resides at the active enhancers together with master TFs, and decommission those enhancers during cell differentiation[48]. The mechanism keeping LSD1 out of function until receiving proper stimulus remains elusive. In classically polarized macrophages, we demonstrated that ZMYND8-mediated LLPS is required to activate LSD1 and turns off SEs with extraordinary efficiency, which again emphasizes that LLPS-regulated gene expression is dependent on the fluidity of phase-separated liquid compartments. Interestingly, when LSD1 fused with p65 and ZMYND8-IDR in vitro, the liquid compartments became smaller than p65/ZMYND8-IDR only (Supplementary Fig. 8c, d). It indicates the decrease of the fluidity of the complex and may be critical for the function of LSD1 demethylation. Another H3K4 demethylase, KDM5C, may also participate in the ZMYND8-mediated SEs regulation during macrophage polarization. Therefore, it is worthwhile to evaluate whether the

H3K4me2 and H3K4me3 pattern regulated by ZMYND8/KDM5C is through LLPS or not in future studies.

Together, we elucidate an epigenetic remodeling mechanism by which epigenetic regulators employ the LLPS mechanism to regulate SEs and related inflammatory responses.

## Methods

**Mouse strains.** The *Zmynd8^flox/flox* mouse model was established by gene targeting via homologous recombination in C57BL/6 embryonic stem cells (Cygene, Guangzhou, China). LoxP sequences were introduced to flank exons 6 and 7 of the *Zmynd8* gene. *Zmynd8^flox/flox* mice were crossed with specific Cre transgenic mice (heterozygous for *Lyz2*-Cre allele) to achieve lineage-specific gene deletion. *Lyz2*-Cre (also known as LysM-Cre) mice on a C57BL/6 background were purchased from the Jackson Laboratory. Gender and age-matched littermates WT *Zmynd8^fl/fl* mice were used as control. All mice were maintained and bred in the pathogen-free facility. Mouse experimental protocols were approved by the Institutional biomedical research ethics committee of Shanghai Institute of Nutrition and Health Sciences, Chinese Academy of Sciences.

**Inflammatory mouse models.** Dextran sodium sulfate (DSS) (MW 36–50 kDa, MP Biomedicals) dissolved in drinking water with 3.5% concentration (w/v) was used to feed six-week-old female WT *Zmynd8^fl/fl* and *Zmynd8* cKO mice. The untreated mice were treated only with water. Mice have been treated for two cycles of DSS at 4 days/cycle and 4 days of pure drinking water between each cycle. In the duration of DSS treatment, to control body weight, general health condition, stool consistency, and fecal blood, mice were monitored and recorded daily. If each mouse had lost more than 25% of body weight, they were euthanized per guidelines.

Six-week-old female mice were fed with an HFD (D12492, 60% of calories from fat Research Diets) for 16 weeks. Body weight was measured weekly. Oral GTTs (1 g/kg) were performed after an overnight fast. For ITTs, mice fasted for 4 h before injecting human regular insulin (1 units/kg). The glucometer monitored Tail's blood glucose levels at 0, 15, 30, 60, 90, and 120 min time points. All animal protocols were approved by the Institutional biomedical research ethics committee of Shanghai Institute of Nutrition and Health Sciences, Chinese Academy of Sciences..

**Plasmids, cells, and reagents.** ZMYND8 construct was a kind gift from Dr. Fei Lan's lab, Fudan University. The IDR region of ZMYND8 between 430 aa and 875 aa was cloned into pET28a with C-terminally GFP-tag for the in vitro protein expression and purification. The IDR region (between 388 aa and 899 aa) was also cloned into pEGFP-N1 to generate GFP–ZMYND8-IDR construct used in HeLa cells. pcDNA3-p300-HA plasmid was a gift from Dr. Yichuan Xiao's lab (SINH, CAS, Shanghai, China). All plasmid constructs were confirmed by Sanger sequencing. All the cloning primers and gRNA sequences are listed in Supplemental Data 1.

Raw264.7, HeLa, MCF7, and 293 T cells were cultured in DMEM medium (Thermo Fisher) supplemented with 10% FBS, 1% penicillin/streptomycin, 2 mM L-glutamine (GIBCO), and 0.1 mM β-mercaptoethanol (Sigma). *Zmynd8* and *p65* KOs were generated by a lentiviral-gRNA-based approach using the LentiCrispr-V2 vector. Mouse BMDMs were generated from WT and *Zmynd8* cKO mice. Bone

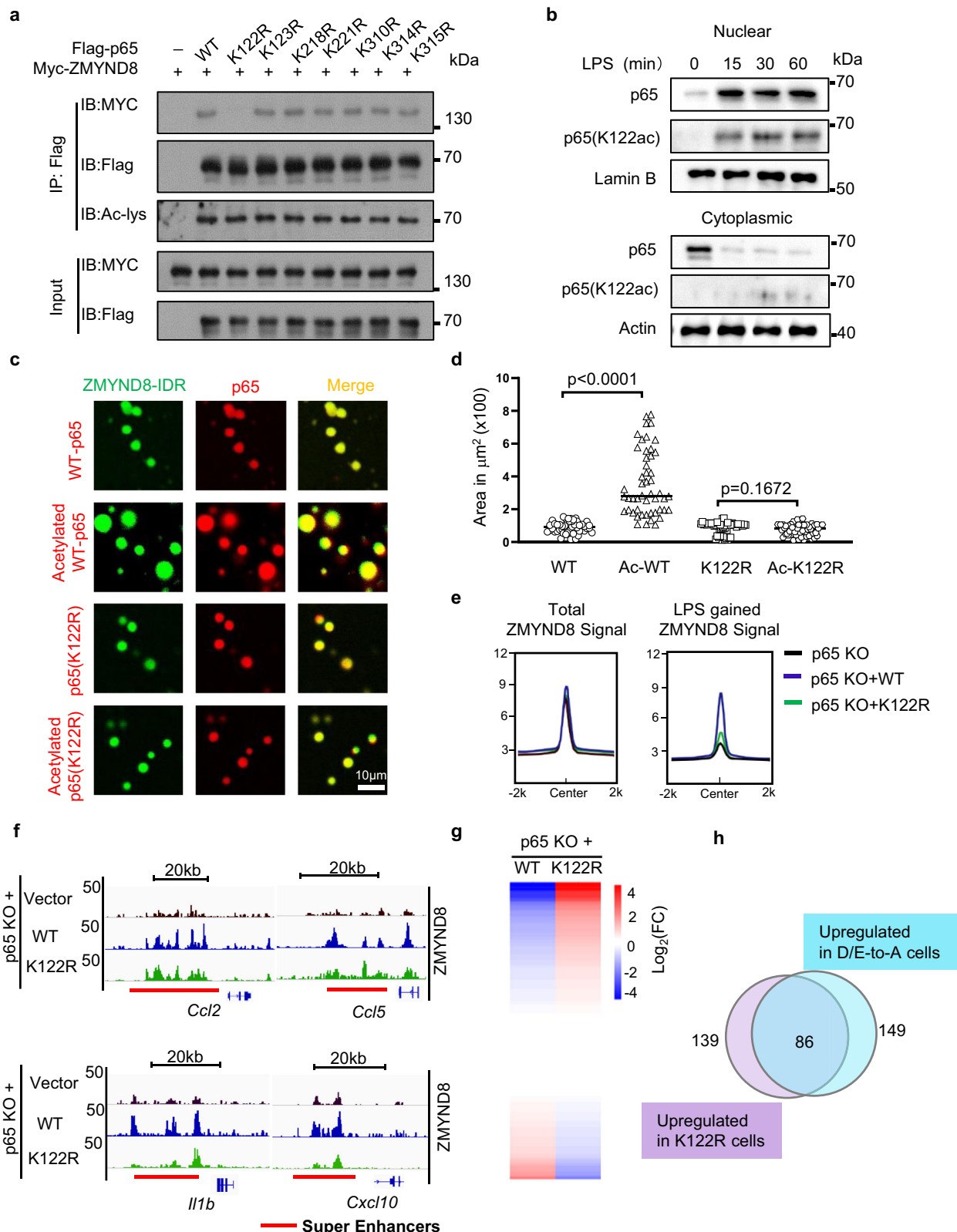

marrow cells from femurs and tibias were isolated and plated at a density of $1 \times 10^6$ to $2 \times 10^6$ cells/ml in RPMI-1640 medium supplemented with 10% FBS (GIBCO) and mM-CSF (20 ng/ml; Novoprotein) on a 10 cm Petri dishes to differentiate for 5 days.

**Flow cytometry**. Single-cell suspensions were obtained from mouse BMs, and spleens were treated with RBC lysis buffer (eBioscience). For surface marker staining, cells were blocked with Fc receptor antibody (catalog no.101310, BioLegend, 1:500) for 10 min at 4 °C before staining with fluorochrome-conjugated surface antibodies: CD11b (clone:M1/70, Biolegend, catalog no.101251, 1:300), F4/80 (clone:BM8, eBioscience, catalog no.17-4801-80, 1:300), Gr-1 (clone:RB6-8C5, eBioscience, catalog no.25-5931-81, 1:300), CD11c (clone: N418, Invitrogen, catalog no.11-0114-85, 1:300). Flow cytometric analysis was performed by using a Gallios flow cytometer (Beckman Coulter). Data were analyzed with FlowJo software (Tree Star) (Version 10).

**Fig. 6 LPS-induced p65 acetylation at the K122 residue directs the docking of ZMYND8 liquid condensates onto NF-κB associated SEs. a** Co-IP assay between Myc-tagged ZMYND8 and various p65 acetylation site mutants in 293 T cells followed by western blot. Experiments were repeated three times. **b** The endogenous K122 acetylation level in the cytosol and nucleus of BMDMs was evaluated by western blot at the indicated time after LPS treatment. Experiments were repeated three times. **c** In vitro droplet formation by incubating GFP-ZMYND8-IDR with mCherry-p65, mCherry-p65-acetylated, mCherry-p65 (K122R), and mCherry-p65 (K122R)-acetylated respectively. Representative images were shown. **d** Statistical analysis of droplet size formed by GFP-ZMYND8-IDR with mCherry-p65, mCherry-p65-acetylated, mCherry-p65 (K122R), and mCherry-p65 (K122R)-acetylated respectively ($n = 50$, examined over three independent experiments). Two-tailed Student's $t$-test determined $p$ values. **e** Genome-wide ChIP-Seq analysis of ZMYND8-binding patterns after LPS treatment in different Raw264.7 cells, including p65 KO cell transduced with Vector (p65 KO), p65 KO transduced with WT p65 (p65 KO + WT p65), and p65 KO transduced with K122R mutant (p65 KO + K122R). The average signal intensity of total ZMYND8 peaks (left) and LPS-gained ZMYND8 peaks (right) was shown separately. **f** Genome browser tracks show that LPS-gained ZMYND8 peaks were recruited to the indicated SE regions in the p65 KO, p65 KO + p65 WT, and p65 KO + K122R Raw264.7 cells after LPS treatment. **g** Heatmap analysis of RNA-Seq results represent differentially expressed genes regulated by ZMYND8 enriched, latent SEs (from Fig. 2d) in p65 KO + WT p65, p65 KO + K122R Raw264.7 cells after LPS treatment. **h** The latent SEs regulated genes enhanced in K122R cells are overlapped with latent SE genes upregulated in D/E-to-A cells. Source data are provided as a Source Data file.

**RNA extraction and quantitative real-time PCR**. Total RNA of collected tissues were extracted with TRIzol RNA Isolation Reagents (Thermo-Invitrogen) and reverse-transcribed with HiScript II Q RT SuperMix for qPCR (Vazyme). Real-time PCR was performed with FastStart Universal SYBR Green Master (Roche) and with different primer sets on QuantStudio 7 Flex Real Real-Time PCR System (Applied Biosystems). The primers used for real-time PCR are shown in Supplementary Data 1. The 2–ΔΔCt method was used to calculate relative expression changes.

**Immunoprecipitation and western blot**. Cells were washed with cold PBS and then lysed with IP buffer (20 mM Tris-HCl, pH 8.0, 150 mM NaCl, 1 mM EDTA, 0.5% NP-40, and 1 mM DTT), plus 1× protease and phosphatase inhibitors (Roch). After centrifuge, soluble protein concentrations were determined using the BCA kit (Pierce). Proteins were applied to SDS-PAGE and transferred to nitrocellulose membranes, followed with primary antibodies incubation overnight at 4 °C, and HRP-conjugated secondary antibody (Cell Signaling Technology) for 2 h at room temperature. Chemiluminescence was determined using the Super Signal West Dura detection system (Thermo Scientific). Proteins were visualized and quantitated by the Bio-Rad ChemiDoc XRS system.

Total lysates were incubated with 2 μg of antibody and protein A/G Sepharose beads (Santa Cruz) overnight at 4 °C. The immunocomplexes were then washed with IP buffer three times and analyzed by SDS-PAGE. ZMYND antibodies used in the immunoprecipitation were purchased from Bethyl (A302-090A, 2 ug/sample). ZMYND antibodies used in the immunoblotting were purchased from Sigma (HPA020949, 1:1000). RelA (p65) antibody (Santa Cruz, catalog no. sc-8008, 1:1000), MYC tag antibody (Santa Cruz, catalog no. sc-40 HRP, 1:2000), FLAG tag antibody (Sigma, catalog no. B3111, 1:1000), Actin (Santa Cruz Biotechnology, catalog no. sc-8432, 1:2000), Lamin B (Santa Cruz Biotechnology, catalog no. sc-374015, 1:1000), LSD1 (Abcam, catalog no. ab129195, 1:1000), His tag antibody (Santa Cruz, catalog no. sc-8036, 1:1000), HA tag antibody (Santa Cruz, catalog no. sc-7392, 1:1000), Acetylated-Lysine antibody (Cell Signaling Technology, catalog no. 9441s, 1:1000), GFP antibody (Cell Signaling Technology, catalog no. 2555s, 1:1000), HDAC1 (Santa Cruz, catalog no. sc-81598, 1:1000), PU.1 (Santa Cruz, catalog no. sc-390405, 1:1000), KDM5C (Santa Cruz, catalog no. sc-376255, 1:1000), p38 MAPK (D13E1) antibody (Cell Signaling Technology, catalog no. 8690s, 1:1000), Phospho-p38 MAPK (Thr180/Tyr182) (D3F9) (Cell Signaling Technology, catalog no.4511s, 1:1000), p44/42 MAPK (Erk1/2) (137F5) (Cell Signaling Technology, catalog no. 4695s, 1:1000), Phospho-p44/42 MAPK (Erk1/2) (Thr202/Tyr204) (D13.14.4E) (Cell Signaling Technology, catalog no. 4370s, 1:1000), SAPK/JNK antibody (Cell Signaling Technology, catalog no. 9252s, 1:1000), Phospho-SAPK/JNK (Thr183/Tyr185) (G9) (Cell Signaling Technology, catalog no. 9255s, 1:1000), Phospho-NF-κB p65 (Ser536) (93H1) (Cell Signaling Technology, catalog no. 3033s, 1:1000), Peroxidase AffiniPure Goat Anti-Rabbit IgG (H + L) (Jackson ImmunoResearch, catalog no. 111-035-003, 1:5000), Peroxidase AffiniPure Goat Anti-Mouse IgG (H + L) (Jackson ImmunoResearch, catalog no. 115-035-003). K122-acetylated p65 antibody was produced by Shanghai Hui Ou Biotechnologies Company.

**RNA-Seq**. In vitro-differentiated BMDMs from WT and *Zmynd8* cKO mice were applied to LPS (100 ng/ml, Sigma) or IL-4 (20 ng/ml, Novoprotein) for 6 h, respectively. Then, total RNA was isolated from those treated cells and untreated cells as well. Three biological replicates were mixed and provided for library construction and sequencing (RiboBio. Ltd). Briefly, libraries were constructed from polyadenylated RNAs and sequenced with an Illumina HiSeq 4000 on an SR-50 run aiming for 30 million reads per sample. Reads were aligned to the mm10 mouse transcriptome using TopHat. Significant genes were defined by the $p$ value and false discovery rate of the cutoff of 0.05 and fold changes ≥1.2, and differential gene expression was subsequently analyzed using the DAVID bioinformatics platform and determined using Cuffdiff. GSEA was performed using software

available from the Broad Institute (www.broadinstitute.org/gsea). Heatmap analyses of RNA Sequencing data were performed using RStudio.

**ChIP, ChIP-Seq, and ATAC-seq**. The ChIP assay procedure was modified from the manufacturer's instructions (EZ-ChIP, millipore). Briefly, isolated BMDMs or mouse Raw264.7 cells (about $1 \times 10^7$ cells) were fixed with 1% methanol-free formaldehyde (Thermo Fisher Scientific) at room temperature for 10 min, followed by quenching with 125 mM glycine. All subsequent steps were implemented as the standard ChIP protocols[42]. And libraries were sequenced with an Hiseq 2500. The FASTQ data were mapped to the mouse genome (mm10) using Bowtie, and significant enrichments were identified by MACS2.0 using Broad Peak mode with $p$ value ≤$1 \times 10^{-5}$, FDR ≤ 0.01 as a cutoff to call peaks from the aligned results. Reads-per-million-normalized wiggle files were displayed in the UCSC genome browser. Antibodies used in ChIP are ZMYND8 (Bethyl, catalog no. A302-090A, 2 μg/sample), H3K4me1 (Cell Signaling Technology, catalog no. 9751, 2 μg/sample), LSD1 (Abcam, catalog no. ab129195, 2 μg/sample), H3K4me2 (Abcam, catalog no. ab32356, 2 ug/sample), p65 (Santa Cruz, catalog no. sc-8008, 2 ug/sample). Shanghai DIATRE Biological Technology performed the ATAC assay. And libraries were sequenced with NovaSeq 6000.

**Super-enhancer identification**. Super-enhancers were identified using ROSE (https://bitbucket.org/young_computation/), which is an implementation of the algorithm. Briefly, this algorithm stitches constituent enhancers together if they are within a certain distance and ranks the enhancers by their input-subtracted signal of H3K27ac. It then separates super-enhancers from typical enhancers by identifying an inflection point of the H3K27ac signal versus enhancer rank. In our study, ROSE was run with a stitching distance of 12500 bp, and then the total ChIP-seq occupancy of H3K27ac[42] from NT and classically polarized BMDMs were applied to the analysis. Meanwhile, increasing ChIP-Seq occupancy of ZMYND8 was ranked based on ChIP-seq of H3K27ac in BMDMs[42], which revealed a clear inflection point in the distribution of ZMYND8 at enhancers. The inflection point was geometrically defined and used to establish the cutoff for super-enhancers.

**FRAP assay and analysis in live cells**. HeLa cells were transfected with various GFP-tagged plasmids by using Lipofectamine 2000 (Thermo). Twenty-four hours later, transfected cells were applied to a live-cell image system (Carl Zeiss LSM880). For EGFP excitation, a 488 nm line of an Argon Krypton laser and fluorescence emission was collected between 500 and 560 nm. A pre-bleach image was captured by averaging four consecutive images. A region of interest (ROI) spot inside the cell was bleached with a laser pulse, lasting between 0.1 and 0.5 s at 100% power without scanning. Then single section images were then collected at 5-s intervals, and at this time, the laser power was typically attenuated to 10% of the maximum. Plots generated from the background were subtracted from the ROI images using the ImageJ 2.1.0 (http://rsb.info.nih.gov/ij).

**Immunofluorescence and DNA FISH**. For immunofluorescence, 4% paraformaldehyde-fixed BMDMs or Raw264.7 cells were first incubated with PBS containing 0.5% Triton X-100 for 15 min at room temperature. Primary antibodies for endogenous ZMYND8 (Sigma, HPA020949) were used at a dilution of 1:200 for 1 h at 37 °C, then washed three times in PBST (PBS containing 0.1% Triton X-100), followed with appropriate fluorescent-conjugated secondary antibody (1:400) dilution for 30 min at room temperature. Goat anti-Rabbit IgG (H + L) Cross-Adsorbed Secondary Antibody, Alexa Fluor 488 (Invitrogen, catalog no. A-11008), Goat anti-Mouse IgG (H + L) Cross-Adsorbed Secondary Antibody, Alexa Fluor 594 (Invitrogen, catalog no. A-11005). Then cells were washed three times with PBST and were ready for confocal analysis. For immunofluorescence with DNA FISH assay, antibodies labeled cells were fixed for a second time with freshly prepared 2% paraformaldehyde for 10 min at room temperature and then treated

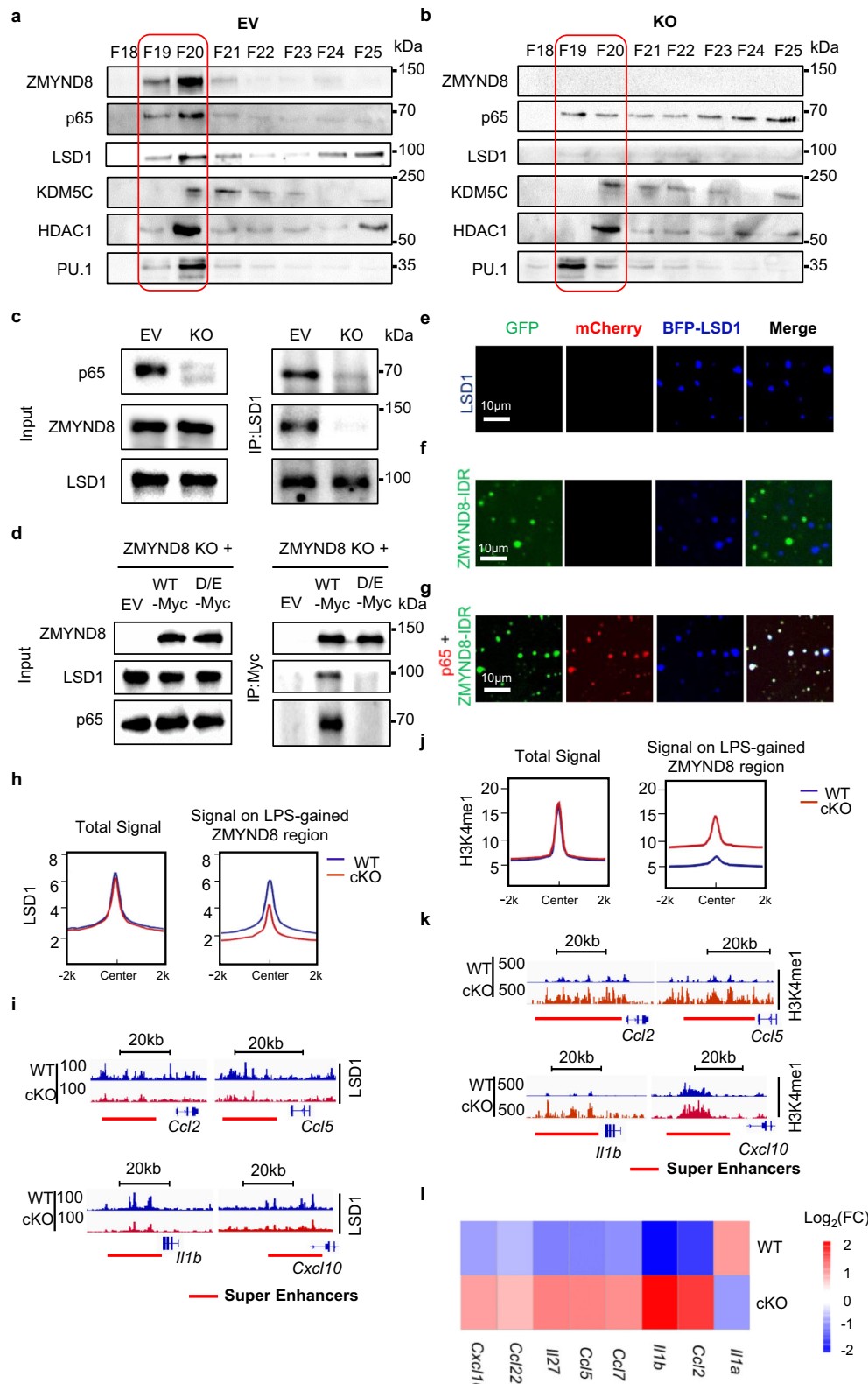

with 0.1 M Tris-HCl (pH 7.4) for 5 min. Before hybridization, coverslips with cells were incubated in 0.1 N HCl for 5 min and washed twice with PBS. Coverslips were treated with RNase A for 1 h at 37 °C, followed by equilibration in 50% formamide/2× SSC for 1 h. Then, probes in the hybridization buffer were applied to the coverslip. Coverslips were heated to 80 °C for 6 min followed by overnight hybridization at 37 °C in a humidified dark chamber. The coverslips were then washed twice with pre-warmed buffer containing 50% formamide/2× SSC and twice with 2× SSC before being finally mounted with Vectashield antifade mounting medium with 4,6-diamidino-2-phenylindole (Vector Laboratories).

DNA FISH probes were custom designed and generated by Shanghai Shinegene Molecular Biotechnology Company to target *Ccl2* super-enhancer. DNA probe sequences for *Ccl2*:

5′-Biotin TGAGGTATTCTGGATTGGACTATTGGCATCAGGGTTG.

**Protein purification**. pET plasmids encoding the IDR region of ZMYND8 or other genes of interest were engineered to include a 6× His tag followed by either mEGFP or mCherry. Plasmids were then transformed into Rosetta-competent cells

**Fig. 7 ZMYND8 liquid condensates activate LSD1 to decommission SEs. a**, **b** Gel filtration chromatography and western blots analyze the indicated protein level in the nuclear fractions purified from control (EV) and *Zmynd8* KO Raw264.7 cells. **c** Endogenous co-IP by using the anti-LSD1 antibody in EV and p65 KO Raw264.7 cells. **d** Co-IP with anti-Myc tag antibody evaluated the interaction between endogenous LSD1 and WT or mutated ZMYND8 (Myc tagged) in different Raw264.7 cells, including *Zmynd8* KO rescued with Vector (*Zmynd8* KO + Vector), WT *Zmynd8* (*Zmynd8* KO + WT *Zmynd8*), and D/E-A mutant (*Zmynd8* KO + D/E-A mutant) Raw264.7 cells. **e**–**g** Representative images of droplet formation containing the indicated proteins, such as BFP-LSD1 only (**e**), GFP-ZMYND8-IDR with BFP-LSD1 (**f**), GFP-ZMYND8-IDR, mCherry-p65, and BFP-LSD1 (**g**). All the experiments above were repeated three times. **h** Genome-wide LSD1-binding patterns in the WT and *Zmynd8* cKO BMDMs after LPS treatment were evaluated by ChIP-Seq. Histogram plot showing the average distribution of total (left) and LSD1 signals at the LPS-gained ZMYND8 peaks region (right). All ChIP-seq signals are displayed from −2 to +2 kb surrounding the center of each annotated LSD1 peak. **i** Genome Browser tracks show the LSD1-binding signals at the selected SE regions in LPS polarized WT and *Zmynd8* cKO BMDMs. **j** Genome-wide H3K4me1 levels in the WT and *Zmynd8* cKO BMDMs after LPS treatment were evaluated by ChIP-Seq. Histogram plot showing the average distribution of total H3K4me1 (left) and H3K4me1 signals at the LPS-gained ZMYND8 peaks region (right). All ChIP-seq signals are displayed from −2 to +2 kb surrounding the center of each annotated H3K4me1 peak. **k** Genome Browser tracks showing the H3K4me1 signals at the selected SE regions in WT and *Zmynd8* cKO BMDMs. **l** Nascent RNA in LPS polarized WT and *Zmynd8* cKO BMDMs were collected, and then the enhancer RNA transcription level near the indicated pro-inflammatory genes was measured by qRT-PCR. Source data are provided as a Source Data file.

(Novoprotein). A single bacterial colony was inoculated into LB media and grown overnight at 37 °C. Then cells containing the constructs were diluted 1:30 in 500 ml LB media and grown at 37 °C for 1.5 h before IPTG addition and for 18 h at 16 °C after 200uM addition of IPTG. Cell pellets of 500 ml of cell culture were resuspended in 15 ml of cell suspension buffer (50 mM Tris 7.5, 300 mM NaCl, 20 mM imidazole) with cOmplete protease inhibitors (Roche) and sonicated (20 cycles of 10 s on, the 20 s off). The lysates were cleared by centrifugation at 12,000g for 30 min, and the supernatant was added to 1 ml of Ni-NTA agarose (Smart Life-sciences). Tubes containing this agarose lysate slurry were rotated for 30 min. The slurry was washed with ten volumes of the lysis buffer containing 50 mM imidazole and eluted with elution buffer containing 250 mM imidazole. The eluted fraction was sent to SDS-PAGE analysis for the right size and replacements with Buffer D (50 mM Tris-HCl pH 7.5, 10% glycerol, and 1 mM DTT).

**In vitro LLPS assay**. Recombinant GFP or mCherry tagged proteins were purified and concentrated to a proper concentration using ultracentrifugal filters (10K MW, Millipore). Recombinant proteins at different concentrations were then added to the LLPS formation buffer containing 50 mM Tris-HCl pH 7.5, 10% glycerol, 1 mM DTT,125 mM NaCl, and 10% PEG8000 as the crowding agent. The protein solution was loaded onto a glass-bottom 96-well plate and then imaged with a confocal microscope with a ×40 objective. Meantime, the various concentrations of salt (NaCl) or 1,6-hexanediol were added into LLPS buffer with the recombinant protein of interest and imaged as described above in the FRAP assay.

**In vitro protein acetylation assay**. The p300 protein was expressed and purified from plasmids transfected HEK-293 T cells. The p65-cherry protein acetylation assay was performed as previously[49]. In the standard assay, a reaction mixture containing 120 μg of purified His-tagged p65 fusion protein, and purified HA-tagged p300 protein from 293 T into 30 μl of acetylation assay buffer (250 mM Tris-HCl, pH 8.0, 50% glycerol, 0.5 mM EDTA, 5 mM dithiothreitol). The reaction mixture was kept in a 30 °C shaking incubator for 4 h. SDS-PAGE evaluated acetylated p65 followed with immunoblot and then added into in vitro LLPS buffer together with other proteins.

**Enhancer RNA (eRNA) assay**. According to the manufacturer's guidelines (Cell-Light TM EU Nascent RNA Capture Kit; RiboBio), nascent RNA was labeled in BMDMs by ethynyl-labeled uridine (EU). Subsequently, the resulting EU-labeled RNA was detected via Cu(I)-catalyzed click chemistry that introduced a Biotin tag for RNA purification. At last, streptavidin-purified RNA was applied to reverse transcriptase-mediated cDNA synthesis and further qPCR analysis.

**Statistical analysis**. One- or two-way ANOVA, where applicable, was performed to determine whether an overall statistically significant change existed before the Student's t-test was used to analyze the difference between any two groups. Data are presented as means ± SD. $P < 0.05$ was considered statistically significant. Analyses were performed with GraphPad Prism(ver.8) statistical software.

**Reporting summary**. Further information on research design is available in the Nature Research Reporting Summary linked to this article.

## Data availability
The data that support this study are available from the corresponding authors upon reasonable request. The next-generation sequencing data generated in this study have been deposited to the GEO database under accession number GSE169470. Public data sets used in this study include GSE3365, GSE81259, GSE55098, and GSE97779. Source data are provided with this paper.

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

## Acknowledgements

We would like to thank Dr. Zhenggang Liu at NCI, Dr. Stefan Muljo at NIAID, and Dr. Jian Cao at Retgus University for helpful suggestions and discussions. This work was supported by China's National Key R&D Program (2018YFA0800602 and 2018YFA0902703) and Research Funding via the National Natural Science Foundation of China (31771576, 81830078, 82050001, and 82071868 to W.L.; 82071866 to Y. Zhang; 81825018, 82130085 to J.Q.).

## Author contributions

P.J., X.L., and X.W. carried out most of the experiments and analyzed the data. L.Y. and Y.X. did the FISH and immunofluorescence experiments. Y.H., W.X., Z.H., and Q.Z. did the mouse BMDM culture and flow cytometry. Y.D., Y. Zang., and M.Z. provided discussions and advice on data analysis. Y. Zhang., J.Q., and W.L. co-directed all the experiments and wrote the manuscript.

## Competing interests

The authors declare no competing interests.
