## [Peer Review File · Nature Communications]

ZMYND8 mediated liquid condensates spatiotemporally decommission the latent super-enhancers during macrophage polarizationREVIEWER COMMENTS

Reviewer #1 (Remarks to the Author):

Jia et al. describe a novel role for ZMYND8, a chromatin reader protein, in decommissioning super-enhancers during macrophage differentiation. This manuscript represents an important set of findings since most studies about enhancers have been about its activation, but it is not as well understood how enhancers are turned off and/or tuned down. ZMYND8 has been reported to have positive as well as negative effects on transcription, but its function in primary macrophages and in vivo is unknown as far as I know. Furthermore, the authors uncover a novel mechanism of action for ZMYND8 that involves liquid-liquid phase separation (LLPS) which is induced by acetylation of Lysine 122 of the p65 subunit of NFκB which then leads to recruitment of LSD1, an H3K4 di-or mono-methylation demethylase. This seems to be a novel negative-feedback loop. Overall, their tour de force work should be regarded as a major advance that promises to be of broad interest to the fields of epigenetics, cell biology, and immunology.

I think their conclusions are well-supported by data which is clearly presented.

Some minor specific comments:

In Abstract, please fix the phrase: "...since mutations coagulation of MYND8 into solid compartments..." There seems to be a grammatical error here.

Page 3, line 45 – please fix "...to insulate protein complexes unapproachable"

Were WT control mice sex-matched littermates?

Based on Methods, authors only analyzed female mice? Did authors analyze male mice in case there are sex-specific differences? Indeed, the immune system of males vs females seems to be different. In case they had noticed anything, it would be worth reporting.

Since Lyz2-Cre is a "knock-in/knockout allele" did authors make sure that they used heterozygous mice for experimental and control mice? Also, since Cre activity may have unanticipated consequences (eg. DNA breaks may affect cell state, it is advisable to control for it.

In any case, please clearly specify what were used as WT control mice for each experiment.

Were plasmids confirmed by sequencing? Primers used for cloning should be listed in Methods.

The gRNAs used in the LentiCrispr-V2 vector should be listed in Methods.

"The p300/CBP protein was expresses and purified from plasmids transfected HEK-293T cells." Source of plasmids should be specified, and "expresses" should be expressed.

Reviewer #2 (Remarks to the Author):

In this work by Jia et al, the authors focused on the function and mechanism of ZMYND8 in regulating LPS induced proinflammatory gene activation in macrophage. Overactivation of proinflammatory genes upon LPS retreatment and interesting phenotypes of ZMYND8 cKO cells and animals were reported. Mechanistically, the authors claimed that the LLPS ability of ZMYND8 was critical for the connection with p65 and LSD1, and the suppression of LPS gained enhancers. Overall, the findings are of interest and importance by revealing the LLPS ability of ZMYND8-LSD1, an enhancer regulatory complex, and establishing the p65-ZMYND8-LSD1 axis in macrophage biology.

For the mechanistic part, the authors attempted to establish the following molecular connection among p65 – ZMYND8 LLPS – LSD1 in controlling the K4me1 level at LPS gained SEs. It is a bit ambiguous to put all the factors together, so there were a few pieces of the evidence missing. To strengthen the study, the following issues should be addressed.

Main concerns

1. In Figure 5, the IDR mutant lacked the LLPS ability and could not interact with p65. However, whether the IDR mutant failed to recruit LSD1 (should be done by co-IP and ChIP-seq or ChIP-qPCR) and induce H3K4 methylation changes should be examined. I believe this is a key experiment, as no other evidence in this study provided a direct connection between ZMYND8 LLPS ability and LSD1/K4me1 dynamics at the target enhancers.

2. Figure 7b, it was strange that LSD1 signals were gone in all fractions, even in the smaller molecular weight fractions to the very right. Based on previous reports, LSD1 should form quite stable complex with HDAC1, so called BHC complex, which should be unlikely affected by ZMYND8. Therefore, the input levels of LSD1 in the control and cKO cells should be examined and provided. If the input levels are comparable, then it would be important to re-examine each fractionation for LSD1. The biochemistry fractionation usually causes sample dilution, so LSD1 level in each fraction could be below the detection limit. But the data in the current version suggested that the whole LSD1 protein level might be reduced in cKO cells, which could be wrong or misleading without necessary control experiments.

3. The authors found that upon ZMYND8 or LSD1 depletion/inhibition, the LPS induced genes were expressed at higher levels compared to the control cells. In Figure 7i, H3K4me1 levels at the LPS-gained ZMYND8 regions were monitored between LPS-treated wt and cKO cells. This is good and consistent with the authors' hypothesis, however, it is not a complete view. H3K4me2 should be included in the examination as well, as LSD1 is an H3K4me2 demethylase. In addition, both H3K4me1 and H3K4me2 ChIP-seq should be performed in the LPS untreated wt and cKO cells to depict the whole view of H3K4 methylation dynamics in the process.

Minor points

ZMYND8 was previously reported to control enhancer H3K4me3 through KDM5C, which was also cited in this study. What the authors observed here, e.g. LPS induced gene were subjected to overly activation in cKO cells, could also be possibly due to loss of KDM5C recruitment and mis-regulation of H3K4me3. I understand that the authors would like to focus on the LLPS, as ZMYND8 and LSD1 could form liquid droplets in the presence of p65. However, this other possibility should at least be mentioned and discussed. Especially, H3K4me1 is just a mark indicating that an enhancer is not silenced, but does not indicate how active an enhancer is. Therefore, a reasonable speculation is that not only H3K4me1 was increased, but H3K4me2 and/or H3K4me3 could also be elevated upon loss of ZMYND8.

Reviewer #3 (Remarks to the Author):

In this study, Jia et al show that an epigenetic reader ZMYND8 forms liquid compartment with NF- κ B/p65 to silence latent super-enhancers and restricts the macrophage-mediated inflammation. Mouse *Zmynd8* suppresses proinflammatory gene expression in LPS-induced M1 BMDMs, *Zmynd8* deficiency aggravates the macrophage-mediated inflammatory responses. ChIP-seq of *Zmynd8* and enhancer markers together with ATAC-seq analysis show that *Zmynd8* preferably redistributes to latent super enhancers (SEs) in LPS polarized macrophages. Mechanistically, ZMYND8 forms liquid-liquid phase-separated (LLPS) condensates in the nucleus. The LLPS is mediated by its IDR region, mutations of the acidic amino acids in the IDR of ZMYND8 compromise the liquidity feature of ZMYND8 and its recruitment onto SEs. Furthermore, they also show that signal-induced p65

acetylation guides the redistribution of ZMYND8 onto NF- κ B associated SEs, and ZMYND8 liquid condensates activate LSD1 to decommission SEs. Overall, this is a comprehensive study showing that ZMYND8 controls the magnitude of immune response through a LLPS model of spatiotemporal transcription control. Most experiments are well executed and conclusions are supported by experimental evidences. A few concerns are listed below.

Major concerns:

1. The proposed model is that acetylation of p65 on K122 recruits ZMYND8 onto SEs. Although they show that single mutation of K122R abolishes p65 binding to ZMYND8, but this does not demonstrate the binding is acetylation dependent. The authors need to perform orthogonal biochemical assays using acetylated proteins and peptides to determine whether the binding is acetylation dependent. Also, which region in ZMYND8 is responsible for the acetylation binding? Since bromodomain is a general acetylation binding motif, it would be interesting to test with full-length protein to see if it is through the bromodomain of ZMYND8. Mutations of the cognate acetyl-binding residues in ZMYND8 should be used to examine their impact on ZMYND8 chromatin occupancy and p65 target gene expression etc.
2. p65 K122 acetylation has been shown to play a role in both gene activation and gene repression. How does it specifically recruit ZMYND8 to genes repressed upon LPS treatment but not to genes activated? Careful comparison of ZMYND8 and p65 or acetylated p65 ChIP-seq signals is needed.
3. The authors used published ATAC-seq data and ChIP-seq data sets of enhancer-associated histone marks and p65 to compare with Zmynd8 ChIP-seq data generated in the current study, but all these published data were generated in polarized BMDMs induced by different reagents (KLA) (Ref. 38). In addition, the H3K4me2 ChIP-seq in Ref. 38 was mistakenly cited as H3K4me1 in this manuscript. The authors need to perform all these ChIP-seq (H3K4me1, H3K27ac and p65 ChIP-seq) in their own cells and treatment.
4. Figure 7e shows the distribution patten of ZMYND8-IDR upon 1,6-HD treatment. This need to be done for the full-length ZMYND8 upon 1,6-HD treatment. Adding FISH experiment (as in Fig. 4a) in 1,6-HD treated cells would further strengthen their conclusion.
5. Figure 5e shows ZMYND8-IDR coalesce with p65 (the sizes of p65 plus ZMYND8-IDR droplets are larger than ZMYND8-IDR droplets), but this is not observed in p65+ZMYND8-IDR+LSD1 droplets (Fig. 7d). Why?
6. Do the RNA or protein levels of ZMYND8 change upon LPS induction, 1,6-HD treatment or in p65 KO cells? Western blots are also needed to show equal expression levels of WT and D/E-A mutants of Zmynd8 in rescued cells.

Minor concerns

1. Fig. 1d and f, please provide numbers of mice used in each group for DSS induced colitis.
2. Are the color keys in Fig. 2b and SI Fig. S4d the same? The Zmynd8 peak densities of LPS-gained peaks in the NT sample look similar to the constitutive Zmynd8 peaks. It is not clear how they are determined as de novo peaks.
3. SI Fig. S4a, it would be helpful to include a pie chart of the genomic distribution of different regions.
4. The pattens of endogenous ZMYND8 in Figures 3 and 4 do not look spheric, it would be more propriate to describe them as nuclear speckles.
5. Fig. 4c, NT is used to indicate without LPS treatment or without 1,6-HD treatment? Please clarify.
6. Fig. 5c, please include IF images of ZMYND8 and p65 in untreated cells.
7. SI Fig. S5e, is it mouse or human ZMYND8? There are many acidic amino acid residues in the IDR region of ZMYND8, please provide information about which 19 D or E residues were mutated in figure legends and the Methods and Materials section.

REVIEWER COMMENTS

Reviewer #1 (Remarks to the Author):

Jia et al. describe a novel role for ZMYND8, a chromatin reader protein, in decommissioning super-enhancers during macrophage differentiation. This manuscript represents an important set of findings since most studies about enhancers have been about its activation, but it is not as well understood how enhancers are turned off and/or tuned down. ZMYND8 has been reported to have positive as well as negative effects on transcription, but its function in primary macrophages and in vivo is unknown as far as I know. Furthermore, the authors uncover a novel mechanism of action for ZMYND8 that involves liquid-liquid phase separation (LLPS) which is induced by acetylation of Lysine 122 of the p65 subunit of NF- κ B which then leads to recruitment of LSD1, an H3K4 di-or mono-methylation demethylase. This seems to be a novel negative-feedback loop. Overall, their tour de force work should be regarded as a major advance that promises to be of broad interest to the fields of epigenetics, cell biology, and immunology.

I think their conclusions are well-supported by data which is clearly presented.

Some minor specific comments:

- 1. In Abstract, please fix the phrase: "...since mutations coagulation of MYND8 into solid compartments..." There seems to be a grammatical error here.
Page 3, line 45 – please fix "...to insulate protein complexes unapproachable"*

Answer (A): We would like to thank the review for all the comments and suggestions. We apologize for these grammatical errors in the abstract and manuscript. We corrected those errors and updated them in the revised manuscript.

- 2. Were WT control mice sex-matched littermates?*

A: We applied the gender and age-matched wild-type (WT) *Zmynd8*^{fl/fl} littermates as the control for *Zmynd8* cKO mice. We now added that information into the Methods section as well as figure legends.

- 3. Based on Methods, authors only analyzed female mice? Did authors analyze male mice in case there are sex-specific differences? Indeed, the immune system of males vs females seems to be different. In case they had noticed anything, it would be worth reporting.*

A: We used female mice in our *in vitro* and *in vivo* investigations but did not evaluate the gender differences. We now tested the LPS polarized BMDMs from male WT and *Zmynd8* cKO mice. The increase of pro-inflammatory gene expression was still

observed in BMDMs derived from male cKO mice (Data shown in the figure below), indicating that animal gender may not influence the function of *Zmynd8* in the macrophages.

After 6 hours LPS treatment, the expression of indicated pro-inflammatory genes in male WT and *Zmynd8* cKO BMDMs was analyzed by qRT-PCR.

4. Since *Lyz2-Cre* is a “knock-in/knockout allele” did authors make sure that they used heterozygous mice for experimental and control mice? Also, since *Cre* activity may have unanticipated consequences (eg. DNA breaks may affect cell state, it is advisable to control for it. In any case, please clearly specify what were used as WT control mice for each experiment.

A: We did use heterozygous *Lyz2-Cre* mice with *Zmynd8^{fl/fl}* genetic background as cKO mice and *Zmynd8^{fl/fl}* only as WT control in our experiments. Those mouse information has been updated in the Methods and Materials Section now.

In addition, we confirmed that the lack of one allele of *Lyz2* (Hz) and the presence of *Cre* enzyme in the *Lyz2-Cre* heterozygous mice do not alter the polarization of BMDMs when compared to WT and *Zmynd8^{fl/fl}* control mice (Data shown below).

After 6 hours LPS treatment, the relative expression of indicated proinflammatory genes in *Lyz2-Cre* (Hz) *Zmynd8^{+/-}* and *Zmynd8^{F/F}* BMDMs was analyzed by qRT-PCR.

5. Were plasmids confirmed by sequencing? Primers used for cloning should be listed in Methods. The gRNAs used in the LentiCrispr-V2 vector should be listed in Methods.

A: All the plasmids used in the present studies were verified by Sanger sequencing. The cloning and sequencing primers and the gRNA sequences used for Crispr KO are now listed in the revised manuscript's supplemental material section (Supplementary information, Table s1).

6. *“The p300/CBP protein was expressed and purified from plasmids transfected HEK-293T cells.” Source of plasmids should be specified, and “expresses” should be expressed.*

A: The pcDNA3-p300-HA plasmid was a gift from Dr. Yichuan Xiao’s lab at SNIH, CAS. We apologize for the grammar error and thank the reviewer for the correction. We added the plasmid source in the Methods and Materials section and corrected the sentence in the revised manuscript.

Reviewer #2 (Remarks to the Author):

In this work by Jia et al, the authors focused on the function and mechanism of ZMYND8 in regulating LPS induced proinflammatory gene activation in macrophage. Overactivation of proinflammatory genes upon LPS retreatment and interesting phenotypes of ZMYND8 cKO cells and animals were reported. Mechanistically, the authors claimed that the LLPS ability of ZMYND8 was critical for the connection with p65 and LSD1, and the suppression of LPS gained enhancers. Overall, the findings are of interest and importance by revealing the LLPS ability of ZMYND8-LSD1, an enhancer regulatory complex, and establishing the p65-ZMYND8-LSD1 axis in macrophage biology.

For the mechanistic part, the authors attempted to establish the following molecular connection among p65 – ZMYND8 LLPS – LSD1 in controlling the K4me1 level at LPS gained SEs. It is a bit ambiguous to put all the factors together, so there were a few pieces of the evidence missing. To strengthen the study, the following issues should be addressed.

Main concerns

1. In Figure 5, the IDR mutant lacked the LLPS ability and could not interact with p65. However, whether the IDR mutant failed to recruit LSD1 (should be done by co-IP and ChIP-seq or ChIP-qPCR) and induce H3K4 methylation changes should be examined. I believe this is a key experiment, as no other evidence in this study provided a direct connection between ZMYND8 LLPS ability and LSD1/K4me1 dynamics at the target enhancers.

A: We thank the reviewer for the comments and suggestions. As suggested, we evaluated the interaction between ZMYND8 IDR mutant and LSD1 in ZMYND8

depleted Raw264.7 cells. We found that the endogenous LSD1 failed to bind ZMYND8 when the IDR region was mutated (Data shown below in Fig. a, and updated in Fig. 7d).

Furthermore, the LSD1 ChIP-qPCR results suggested that LSD1 binding on the SEs was attenuated in the IDR mutant cells compared to that in WT cells. Meanwhile, the H3K4 mono-methylation (H3K4me1) and di-methylation (H3K4me2) in those SE regions were upregulated in D/E-A mutant cells (Data shown below in Fig. b). Now, we updated those new data in Supplementary information, Fig. s8k-s8m.

(a) ZMYND8 KO Raw264.7 cells were stably transduced with EV, Myc-ZMYND8, Myc-ZMYND8 mutant (D/E-A). Co-immunoprecipitation with anti-Myc tag antibody evaluated the interaction between endogenous LSD1 and WT or mutated ZMYND8 (Myc tagged).
 (b) ChIP-qPCR analysis of LSD1 binding intensities, as well as H3K4me1 and H3K4me2 levels at SE regions of indicated genes in different Raw264.7 cells (including *Zmynd8* KO, *Zmynd8* KO + WT *Zmynd8*, and *Zmynd8* KO + DE-A mutant cells). RT-qPCR primers targeted SE locations upstream of the TSS were shown in the brackets.

2. Figure 7b, it was strange that LSD1 signals were gone in all fractions, even in the smaller molecular weight fractions to the very right. Based on previous reports, LSD1 should form quite stable complex with HDAC1, so called BHC complex, which should be unlikely affected by ZMYND8. Therefore, the input levels of LSD1 in the control and cKO cells should be examined and provided. If the input levels are comparable, then it would be important to re-examine each fractionation for LSD1. The biochemistry fractionation usually causes sample dilution, so LSD1 level in each fraction could be below the detection limit. But the data in the current version suggested that the whole LSD1 protein level might be reduced in cKO cells, which could be wrong or misleading without necessary control experiments.

A: We apologize for the lacking of input control for the gel-filtration experiments. We

added the western blot results of the input of control and KO samples. The overall LSD1 protein level was equal between WT and *Zmynd8* deficient cells (Shown below in Fig. a, and Supplemental information, Fig. s8a). In the original Fig. 7b, we only presented the high molecular-weight fractions (from #18-25) where ZMYND8 is located in WT cells. Without ZMYND8, LSD1 protein underwent a translocation to fraction #36 to #38, representing a lower molecular-weight complex. HDAC1 was absent in those fractions, but PU.1 was still colocalized with LSD1 (Show below in Fig. b, and Supplemental information, Fig. s8b). In the WT ZMYND8 cells, LSD1 was absent in fractions #36 to #38 (Shown below in Fig. c). These results indicated that without *Zmynd8* in LPS polarized macrophages, LSD1 may disassociate from HDAC1. This disconnection may be cell-specific, and a detailed mechanism needs further investigation.

(a) The input of WT and KO samples for gel filtration chromatography analysis. (b) Western blots analyzed the indicated protein level in different fractions purified from *Zmynd8* KO Raw264.7 cells and WT cells (c).

3. The authors found that upon ZMYND8 or LSD1 depletion/inhibition, the LPS induced genes were expressed at higher levels compared to the control cells. In Figure 7i, H3K4me1 levels at the LPS-gained ZMYND8 regions were monitored between LPS-treated wt and cKO cells. This is good and consistent with the authors' hypothesis, however, it is not a complete view. H3K4me2 should be included in the examination as well, as LSD1 is an H3K4me2 demethylase. In addition, both H3K4me1 and H3K4me2 ChIP-seq should be performed in the LPS untreated wt and cKO cells to depict the whole view of H3K4 methylation dynamics in the process.

A: We fully agree with the reviewer that H3K4 di-methylation (H3K4me2) should be evaluated since LSD1 is also an H3K4me2 demethylase. Now, we finished the ChIP-Seq by using the H3K4me2 antibody. Similar to the H3K4me1 pattern, the overall H3K4me2 level in *Zmynd8* cKO BMDMs is not changed, but the H3K4me2 signal at the LPS-gained ZMYND8 binding region is elevated (Fig. a-b shown below, and Supplementary information, Fig. s8h and s8i). The genome browser demonstrated that the H3K4me2 level in the SE regions of pro-inflammatory genes is elevated (Shown below in Fig. c, and Supplemental information, Fig. s8j).

We also applied H3K4me1 and H3K4me2 ChIP-Seq in untreated WT and cKO cells. As expected, WT and cKO BMDMs showed comparable H3K4me1 and H3K4me2 levels without (NT) and with LPS treatment (LPS) as well (shown below in Fig. d and e, and Supplementary information, Fig. s8e, and s8f). These results indicate that ZMYND8 does not influence the global H3K4 methylation pattern, recapitulating the normal BMDM development results.

(a-b) Heatmap and Histogram analysis of H3K4me2 ChIP-Seq data obtained from LPS-treated WT and *Zmynd8* cKO BMDMs. Peak density heatmap showing total (left) and LPS-gained ZMYND8 binding signals (Right). Each row shows ± 2 kb centered regions of LPS-gained ZMYND8 peaks. (c) Genome Browser tracks show H3K4me2 ChIP-Seq signals in the NT and LPS-treated BMDMs at selected SEs regions. (d) The H3K4me1 and H3K4me2 levels in untreated WT and cKO BMDMs (NT), and LPS treated cells (e) were evaluated by ChIP-Seq.

Minor points

ZMYND8 was previously reported to control enhancer H3K4me3 through *KDM5C*,

which was also cited in this study. What the authors observed here, e.g. LPS induced gene were subjected to overly activation in cKO cells, could also be possibly due to loss of KDM5C recruitment and mis-regulation of H3K4me3. I understand that the authors would like to focus on the LLPS, as ZMYND8 and LSD1 could form liquid droplets in the presence of p65. However, this other possibility should at least be mentioned and discussed. Especially, H3K4me1 is just a mark indicating that an enhancer is not silenced, but does not indicate how active an enhancer is. Therefore, a reasonable speculation is that not only H3K4me1 was increased, but H3K4me2 and/or H3K4me3 could also be elevated upon loss of ZMYND8.

A: We agree with the reviewer that besides LSD1, other epigenetic regulators, such as KDM5C and the H3K4me2 and H3K4me3 levels regulated by KDM5C/ZMYND8, may also involve in macrophages polarization both *in vitro* and *in vivo*. We now discuss that point in the discussion section in our revised manuscript and will apply investigations in our future studies.

To ensure the activation of enhancers we focus on, we also evaluated the transcription of enhancer RNA (eRNA). The eRNA level of several SEs was upregulated in *Zmynd8* cKO cells (Fig. 71), suggesting these enhancer regions are active.

Reviewer #3 (Remarks to the Author):

*In this study, Jia et al show that an epigenetic reader ZMYND8 forms liquid compartment with NF- κ B/p65 to silence latent super-enhancers and restricts the macrophage-mediated inflammation. Mouse *Zmynd8* suppresses proinflammatory gene expression in LPS-induced M1 BMDMs, *Zmynd8* deficiency aggravates the macrophage-mediated inflammatory responses. ChIP-seq of *Zmynd8* and enhancer markers together with ATAC-seq analysis show that *Zmynd8* preferably redistributes to latent super enhancers (SEs) in LPS polarized macrophages. Mechanistically, ZMYND8 forms liquid-liquid phase-separated (LLPS) condensates in the nucleus. The LLPS is mediated by its IDR region, mutations of the acidic amino acids in the IDR of ZMYND8 compromise the liquidity feature of ZMYND8 and its recruitment onto SEs. Furthermore, they also show that signal-induced p65 acetylation guides the redistribution of ZMYND8 onto NF- κ B associated SEs, and ZMYND8 liquid condensates activate LSD1 to decommission SEs. Overall, this is a comprehensive study showing that ZMYND8 controls the magnitude of immune response through a LLPS model of spatiotemporal transcription control. Most experiments are well executed and conclusions are supported by experimental evidences. A few concerns are listed below.*

Major concerns:

- 1. The proposed model is that acetylation of p65 on K122 recruits ZMYND8 onto SEs. Although they show that single mutation of K122R abolishes p65 binding to ZMYND8, but this does not demonstrate the binding is acetylation dependent. The authors need*

to perform orthogonal biochemical assays using acetylated proteins and peptides to determine whether the binding is acetylation dependent. Also, which region in ZMYND8 is responsible for the acetylation binding? Since bromodomain is a general acetylation binding motif, it would be interesting to test with full-length protein to see if it is through the bromodomain of ZMYND8. Mutations of the cognate acetyl-binding residues in ZMYND8 should be used to examine their impact on ZMYND8 chromatin occupancy and p65 target gene expression etc.

A: We thank the reviewer for the helpful suggestions. To confirm the binding of ZMYND8 and p65 is acetylation dependent, we expressed and purified His-tagged WT and K122R mutant p65 proteins from *E.coli* and applied *in vitro* acetylation by incubations with p300 acetyltransferase (Shown in Fig. a below). Then we evaluated the acetylated p65 interaction with ZMYND8 through the *in vitro* His-pull-down assay. The interaction between ZMYND8 and acetylated WT p65 was stronger than un-acetylated WT p65. In contrast, although other acetylation sites on K122R mutant p65 were remained, K122R mutant p65 showed equal binding affinity with ZMYND8 before and after acetylation (Shown in Fig. b below, and supplemental information Fig. s7d).

We also investigated the protein domain of ZMYND8 responsible for the interaction with acetylated p65. In 293T cells, we co-transfected Flag-p65, p300-HA with different ZMYND8 fragment constructs or mutants, respectively. We found the deletion of the IDR domain or MYND domain of ZMYND8 disrupted the interaction with acetylated p65. In contrast, the deletion of the BRD domain or cognate acetyl-binding sites mutations (Y247A/N248A) (*Mol Cell* 63, 470-484) did not influence the interaction between p65 and ZMYND8 (Shown below in Fig. c, and Supplemental information, Fig. s7e). These results indicate that the ZMYND8 recognizes the acetylated p65 through the IDR instead of the BRD domain. However, as a histone acetylation reader, the BRD domain is still critical for the suppressive effect of ZMYND8 on gene expression. When we transduced BRD^{Y247A/N248A} (AA) mutated into *Zmynd8* KO Rwa264.7 cells, we found that AA mutant failed to suppress the pro-inflammatory gene expression compared to WT ZMYND8 (Shown below in Fig. d, and Supplemental information, Fig. s7f). The AA mutant showed less binding on the SEs region of those pro-inflammatory genes (Shown below in Fig. e, and Supplemental information, Fig. s7g). Thereby, p65 acetylation strengthens the association between ZMYND8 and p65, but ZMYND8 recruitment to the chromatin through its BRD domain is a prerequisite.

Acetylated His-p65 in vitro pull-down assay. (a) FLAG-ZMYND8 from transfected HEK293T cells were incubated with non-acetylated or acetylated His-WT p65 and His-p65 K122R mutant proteins. (b) Then His-pull down assay was applied to evaluate the binding of ZMYND8 with p65. (c) Different ZMYND8 fragments and mutants were co-immunoprecipitated with p65 in the presence of p300. (d) After six hours of LPS treatment, the relative expression of indicated proinflammatory genes was evaluated in different Raw264.7 cells, including *Zmynd8* KO, *Zmynd8* KO + WT *Zmynd8*, and *Zmynd8* KO + (Y247A/N248A) AA mutated cells by qRT-PCR. (e) ChIP-qPCR analysis of ZMYND8 binding intensities at indicated SE regions in different Raw264.7 cells, including *Zmynd8* KO, *Zmynd8* KO + WT *Zmynd8*, and *Zmynd8* KO + AA mutated cells. (f) Immunoblot of WT ZMYND8 and AA mutant in different Raw264.7 cells.

2. *p65 K122 acetylation has been shown to play a role in both gene activation and gene repression. How does it specifically recruit ZMYND8 to genes repressed upon LPS treatment but not to genes activated? Careful comparison of ZMYND8 and p65 or acetylated p65 ChIP-seq signals is needed.*

A: We apologize for not clarifying the p65 acetylation-regulated transcription in our initial submission. Rel-A/p65 acetylation plays a crucial role in regulating NF- κ B activation (*Nat Rev Mol Cell Biol* 5, 392–401.). p65 K221, 310 acetylations enhance the transcription of NF- κ B target genes (*EMBO J* 21: 6539–6548.), while K122 and K123 suppress but do not stimulate the p65 transcriptional activity (*J Biol Chem* 278:2758–2766.). We compared the transcriptome between p65-WT and p65-K122R

mutant Raw264.7 cells. Around three-quarters of the differentially expressed genes in K122R cells were upregulated, supporting the previous finding that K122 acetylation is a suppressive modification on p65. We then introduced the ZMYND8 ChIP-Seq and p65 ChIP-Seq results into the analysis. Most upregulated genes are NF- κ B target genes co-occupied by ZMYND8 and p65 together (Fig. a, below).

In contrast, most downregulated genes in K122R cells are not NF- κ B/p65 target genes. p65 or ZMYND8 rarely binds onto the regulatory regions of those genes (Fig. b, below), which indicates that those activated genes by K122 acetylation are not direct targets of p65. Other regulators and pathways downstream of the K122 acetylation are involved.

Unfortunately, our K122 acetylation antibody can not be used for ChIP. A K122 acetylation-specific binding pattern in macrophages is lacking. But based on the RNA-Seq and ChIP-Seq results, we concluded that K122 acetylated p65 recruits ZMYND8 to mainly suppresses NF- κ B target gene expression.

Venn diagram of the overlapped genes associated with ZMYND8, p65 (ChIP-Seq results) and upregulated genes in K122R mutant cells (RNA-seq) (a) or downregulated genes (RNA-seq) in K122R mutant cells (b).

3. *The authors used published ATAC-seq data and ChIP-seq data sets of enhancer-associated histone marks and p65 to compare with Zmynd8 ChIP-seq data generated in the current study, but all these published data were generated in polarized BMDMs induced by different reagents (KLA) (Ref. 38). In addition, the H3K4me2 ChIP-seq in Ref. 38 was mistakenly cited as H3K4me1 in this manuscript. The authors need to perform all these ChIP-seq (H3K4me1, H3K27ac and p65 ChIP-seq) in their own cells and treatment.*

A: We apologized for the mistake when citing the reference data set. As suggested, we now perform all the ChIP-Seq experiments, including H3K4me1, H3K27ac, and p65 ChIP-Seq, in our experimental settings. Since KLA is a TLR4 agonist same as LPS, new sequencing data recapitulates the previous results and supports our original conclusion (Shown below in Fig. a-c). The ATAC-seq data used at the beginning were

performed in-house. We updated new results about H3K4me1 and H3K27ac ChIP-Seq in Fig.2b and 2e; p65 ChIP-Seq results in Fig. 5k and 5l.

We also re-analyzed SE-related genes by using new sequencing results, especially H3K27ac ChIP-Seq data. Subtle differences that occurred. There are 362 genes (378 genes in the previous analysis) associated with LPS induced latent SEs, among which 259 (285 before) are overlapped with LPS-gained ZMYND8 peaks-related genes. As shown previously, most of those overlapped genes are upregulated in cKO BMDMs, and KEGG pathway analysis indicates multiple inflammatory pathways, including NF- κ B. We have updated those results in the revised manuscript and Fig. 2f-2h.

(a) WT BMDMs were treated without (NT) or with (LPS) and then applied to H3K4me1 and H3K27ac ChIP-Seq analysis. H3K4me1 and H3K27ac signal at the specific regions corresponding to LPS-gained ZMYND8 peaks, LPS-lost peaks, and constitutively bound areas were shown. Each row shows ± 2 kb centered regions of ZMYND8 peaks. (b) Genome Browser tracks show ZMYND8, H3K4me1, H3K27ac ChIP-seq signals and chromatin accessibility (ATAC-Seq) in the NT and LPS-treated macrophages at selected SE regions. (c) Heatmap analysis of the colocalization of p65 ChIP-Seq signals with LPS-gained and LPS-lost ZMYND8 peaks.

4. Figure 7e shows the distribution pattern of ZMYND8-IDR upon 1,6-HD treatment. This needs to be done for the full-length ZMYND8 upon 1,6-HD treatment. Adding FISH experiment (as in Fig. 4a) in 1,6-HD treated cells would further strengthen their conclusion.

A: We now expressed full-length (FL) ZMYND8 and applied purified FL-ZMYND8 protein to the LLPS buffer *in vitro*. FL-ZMYND8 also formed spheric liquid condensates in LLPS buffer, and those liquid compartments are 1,6-HD sensitive (Fig. a and b shown below, and Supplemental information, Fig. s5h). We also did the FISH experiment in 1,6-HD treated BMDMs. After treatment, immunofluorescence signals indicate the ZMYND8 compartments are reduced. ZMYND8 puncta and SE probes were fully separated (Fig. c shown below, and Supplemental Information, Fig. s5k).

(a-b) Purified FL-ZMYND8 protein forms spheric liquid compartments in LLPS buffer *in vitro*, and 1,6-HD treatment disperses those droplets significantly. (c) Images of the indicated immunofluorescence of ZMYND8 (Green) and FISH (*Ccl2* probes, Red) in 1,6-HD treated BMDMs are shown, along with the merged channels (overlapping signal in white).

5. Figure 5e shows ZMYND8-IDR coalesce with p65 (the sizes of p65 plus ZMYND8-IDR droplets are larger than ZMYND8-IDR droplets), but this is not observed in p65+ZMYND8-IDR+LSD1 droplets (Fig. 7d). Why?

A: We noticed that when LSD1 coalesced with liquid droplets formed by ZMYND8 and p65 *in vitro*, the fused complex appeared to be smaller with statistical significance after careful calculation (Shown below in Fig. a-b). We evaluated the liquid property of ZMYND8/p65/LSD1 compartments by FRAP assay and found that LSD1 decreases the FRAP recovery efficiency (Fig. c), which indicates that LSD1 decreases the fluidity of the protein complex. This reduced liquid property may benefit for the LSD1/p65/ZMYND8 complex to de-methylate the H3K4 modifications. We added those data in Supplemental information, Fig. s8c and s8d, and updated the discussion

section.

(a) Representative images showed liquid droplets formed by indicated proteins. GFP-ZMYND8-IDR only, GFP-ZMYND8-IDR with mCherry-p65, GFP-ZMYND8-IDR, mCherry-p65, and BFP-LSD1. (b) Droplet size formed by GFP-ZMYND8-IDR only, GFP-ZMYND8-IDR with mCherry-p65, GFP-ZMYND8-IDR, mCherry-p65, and BFP-LSD1, respectively. (c) FRAP efficiency of liquid droplets formed by GFP-ZMYND8-IDR, GFP-IDR+mCherry-p65, GFP-IDR+mCherry-p65+BFP-LSD1, respectively.

6. Do the RNA or protein levels of ZMYND8 change upon LPS induction, 1,6-HD treatment or in p65 KO cells? Western blots are also needed to show equal expression levels of WT and D/E-A mutants of Zmynd8 in rescued cells.

A: The protein level of ZMYND8 is pretty much stable until 24 hours post-LPS-polarization (presented in Fig. a below, and in Supplemental information, Fig. s2h). Zmynd8 mRNA and protein levels are not changed in 1,6-HD treated cells (Shown below in Fig. b and c, and Supplemental information, Fig. s5i and s5j). p65 KO cells also have equal ZMYND8 protein levels compared to EV control cells (Shown in Fig. d below, and Fig. 7c). Meanwhile, the D/E-A mutation does not influence the ZMYND8 protein stability. WT and ZMYND8 IDR mutant cells show equal levels of ZMYND8 protein (Shown below in Fig. e), with another independent supporting evidence in Fig. 7d.

(a) Immunoblotting of ZMYND8 protein level in WT BMDMs after LPS treatment. Relative expression of *Zmynd8* mRNA (c) and protein level (d) in 1,6HD-treated BMDMs treated with or without 1,6-HD. Immunoblotting of ZMYND8 in WT and p65-KO Raw264.7 cells (e) and in *Zmynd8*-KO+ WT and +DE-A mutant Raw264.7 cells (f).

Minor concerns

1. Fig. 1d and f, please provide numbers of mice used in each group for DSS induced colitis.

A: We used five mice in each group for the DSS-induced colitis experiments. Animal experiments were repeated three times. We now updated the exact mouse numbers and other information in the methods and the figure legends.

2. Are the color keys in Fig. 2b and SI Fig. S4d the same? The *Zmynd8* peak densities of LPS-gained peaks in the NT sample look similar to the constitutive *Zmynd8* peaks. It is not clear how they are determined as *de novo* peaks.

A: The color keys are the same in the two figures. Although heatmap peak densities in Fig. 2b and Fig. s4d NT samples look similar, they correspond to different chromatin regions. We apologize for the unclear definition of *de novo* peaks in our manuscript. We defined the newly occupied ZMYND8 peaks on the genome after LPS polarization as *de novo* peaks and referred to LPS-gained ZMYND8 peaks. These peaks locate at the regions that initially lack ZMYND8 binding before the LPS induced classical polarization, as shown in Fig. 2e.

The signal density at the constitutively bound ZMYND8 peaks does not change after LPS treatment, such as the Genome Browser tracks shown below around genes: *Mir704* and *Ezh2*.

Genome browser tracks show that ZMYND8 constitutively bound regions after LPS treatment were shown.

3. SI Fig. S4a, it would be helpful to include a pie chart of the genomic distribution of different regions.

A: We updated the supplemental information Fig. s4a using pie charts with detailed information about genomic distribution.

The updated pie chart shows the detailed distribution of ZMYND8 ChIP-seq peaks on the genome of NT (left) and LPS-treated (right) BMDMs.

4. The patterns of endogenous ZMYND8 in Figures 3 and 4 do not look spheric, it would be more appropriate to describe them as nuclear speckles.

A: We agree with the reviewer that some endogenous ZMYND8 compartments did not look spheric, and we address them nuclear speckles in the manuscript as suggested.

7. Fig. 4c, NT is used to indicate without LPS treatment or without 1,6-HD treatment? Please clarify.

A: NT here indicated cells without 1,6-HD treatment. We now clarify this by using “-1,6 HD” and “+1,6 HD” to avoid confusion in the figures and figure legends.

8. Fig. 5c, please include IF images of ZMYND8 and p65 in untreated cells.

A: We apologize for the missing images of ZMYND8 and p65 in untreated cells. Now we added the typical IF images (below) into the supplemental information Fig. s6a. Before polarization, p65 was mainly in the cytoplasm, and therefore rarely colocalized

with nuclear ZMYND8. The statistic analysis results are already in Fig. 5d.

Immunofluorescence staining shows endogenous ZMYND8 (Green) and endogenous p65 (Red) in mBMDMs before LPS treatment. DAPI indicates the nuclear staining in blue.

9. *SI Fig. S5e, is it mouse or human ZMYND8? There are many acidic amino acid residues in the IDR region of ZMYND8, please provide information about which 19 D or E residues were mutated in figure legends and the Methods and Materials section.*

A: We used human ZMYND8 in our exogenous expression systems. To disrupt the weak multi-valent interaction during the LLPS, we mutated all the D/E acidic amino acids in the IDR region of ZMYND, which is located from amino acids #388 to #899. There should be a total of 66 Aspartic acids (D) and Glutamic acids (E) in this region. We accidentally typed the wrong number, 19 instead of 66, in our manuscript. We feel deeply sorry about this mistake and sincerely thank the reviewer for pointing out this error. We carefully checked our mutant plasmids and cell lines by sequencing again. The sequencing results are also listed below. We also updated that information in the Methods and Materials section and supplemental information Fig. s5e now.

(a) The distribution of amino acid residues Aspartic acid (D) and Glutamic acids (E) in the IDR region of ZMYND8. (b) The sequencing results confirmed the mutation of all 66 Ds and Es in the IDR region of ZMYND8.

REVIEWERS' COMMENTS

Reviewer #1 (Remarks to the Author):

The authors did an exemplary job addressing all the reviewers' concerns.

Reviewer #2 (Remarks to the Author):

The authors have addressed my concern in full, and now the manuscript is ready for acceptance.

Reviewer #3 (Remarks to the Author):

In the revised version of manuscript, the authors have provided quite some new data, which helped to address most of my concerns. These additional data, corrections and revisions have substantially strengthened the manuscript to support their conclusions.

Minor issues:

1. SI Fig. S4a, what I suggested previously is to include a pie chart of general distribution of different elements, such as promoter, exon, intron and intergenic region in the genome. This will give a better view in which elements ZMYND8 is enriched. The pie charts of ZMYND8 peaks detected in NT and LPS samples provided in the original version of SI Fig S4a will be fine, it is not necessary to divide into so detailed categories in current SI Fig. S4a.

2. Additional proofreading will be helpful. For examples: in Fig 3c, PWP should be PWWP; in line 218, "fluorescence situ hybridization" should be "in situ".

REVIEWERS' COMMENTS

Reviewer #1 (Remarks to the Author):

The authors did an exemplary job addressing all the reviewers' concerns.

Answer (A): We would like to thank the reviewer for the helpful suggestions and comments.

Reviewer #2 (Remarks to the Author):

The authors have addressed my concern in full, and now the manuscript is ready for acceptance.

A: We would like to thank the reviewer for the helpful suggestions and comments.

Reviewer #3 (Remarks to the Author):

In the revised version of manuscript, the authors have provided quite some new data, which helped to address most of my concerns. These additional data, corrections and revisions have substantially strengthened the manuscript to support their conclusions.

Minor issues:

1. SI Fig. S4a, what I suggested previously is to include a pie chart of general distribution of different elements, such as promoter, exon, intron and intergenic region in the genome. This will give a better view in which elements ZMYND8 is enriched. The pie charts of ZMYND8 peaks detected in NT and LPS samples provided in the original version of SI Fig S4a will be fine, it is not necessary to divide into so detailed categories in current SI Fig. S4a.

A: We thank the reviewer for the suggestions and corrections. We have replaced the pie chart with the previous version in Supplementary Fig. 4a.

2. Additional proofreading will be helpful. For examples: in Fig 3c, PWP should be PWWP; in line 218, "fluorescence situ hybridization" should be "in situ".

A: We apologize for those remaining mistakes. We now corrected those typos and have gone through the manuscript with careful proof-reading.